# Mitigating Spurious Correlations in Zero-Shot Multimodal Models

**Shenyu Lu, Junyi Chai & Xiaoqian Wang**[*]
Elmore Family School of Electrical and Computer Engineering
Purdue University
West Lafayette, IN 47906, USA
`{lu876,chai28,joywang}@purdue.edu`

## Abstract

Multimodal models or Vision Language Models (VLMs) have reshaped the paradigm in machine learning, offering zero-shot capabilities that require no additional training when adapted to new classification tasks. However, despite their advancements, spurious correlations still exist in VLMs. Existing approaches to tackle this issue often require target label annotations, contradicting the principle of zero-shot classification, or they primarily focus on a single modality, risking misalignment between text and image modalities. Others rely on extensive domain knowledge or large language models (LLMs) to characterize spurious features, making the performance sensitive to the generated prompts and undermining zero-shot capability. In response, we propose a new solution that tackles spurious correlations in VLMs within the zero-shot setting. Our approach utilizes a translation operation that preserves the latent space distribution to address issues of spurious correlations. In particular, our method is grounded in and inspired by a theoretical analysis, which identifies that the optimal translation directions are along the spurious vector. As VLMs unify two modalities, we compute spurious vectors from the text prompts and guide the translation for image embeddings, aligning the requirements for the fusion of different modalities in VLMs. We conducted experiments on benchmark datasets, which have shown significant improvements in worst-group accuracy. Additionally, our visualizations of VLMs further demonstrate the effectiveness of this intervention.

## 1 Introduction

Vision Language Models (VLMs) have significantly enhanced the capabilities of machine learning systems. Contrastive Language-Image Pretraining (CLIP) (Radford et al., 2021), which bridges the fields of computer vision and natural language processing, has profoundly transformed the landscape. One of the fascinating capabilities of VLMs is their zero-shot functionality (Guo et al., 2023). This functionality enables models to infer the most probable answer from a set of potential responses provided by the user, even without training on the specific dataset.

Despite the power of VLMs, these models still suffer from spurious correlations (Zheng et al., 2024; Dehdashtian et al., 2024; Wortsman et al., 2022), a phenomenon where predictions are based on irrelevant features, leading to detrimental performance for certain groups (Sagawa et al., 2019). Spurious correlations pose significant risks in high-stakes settings such as medical diagnostics. For instance, in diagnosing skin cancer, if a color patch is spuriously correlated with benign samples, the model may erroneously base its predictions on the presence of this color patch (Yan et al., 2023; Nauta et al., 2021) (See Figure 1 ISIC Dataset (Codella et al., 2019)).

Addressing spurious correlations in VLMs is increasingly imperative. Efforts such as (Yang et al., 2023; Pang et al., 2024; Goyal et al., 2023; Zhang & Ré, 2022; Wang et al., 2023) have aimed to mitigate spurious correlations issues within VLMs. However, these methods rely on target labels, a practice that contradicts the label-free requirements of zero-shot classification.

---

[*]Corresponding author.

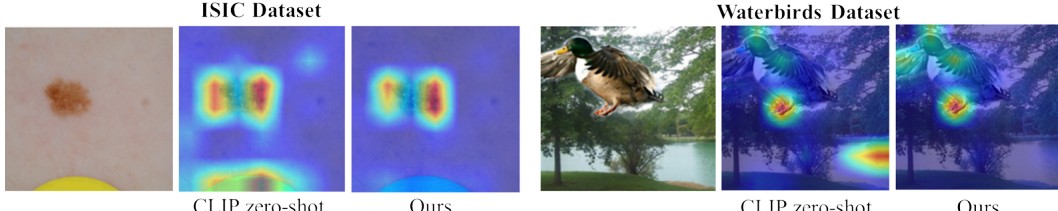

Figure 1: Heatmap visualization for zero-shot classification. The *benign lesion* class in the ISIC dataset is spuriously correlated with the presence of color patches, leading to predictions of benign lesions being dangerously dependent on this feature in the biomedical setting. Similarly, in the Waterbirds dataset, there is a spurious correlation between waterbirds and water backgrounds. Our approach effectively decorrelates these spurious relationships without requiring a training process, promoting group robustness in the zero-shot setting.

A key characteristic of VLMs is the integration of an image encoder and a text encoder, which process image and text inputs, respectively. These inputs are transformed into image embeddings and text embeddings. Many studies (An et al., 2024; Chuang et al., 2023; Trager et al., 2023) have concentrated on mitigating spurious correlations via text embeddings. However, these methods present several challenges. Firstly, they concentrate exclusively on a single modality, posing a substantial risk of misalignment between modalities. This contradicts the principle of matching different modalities in VLMs. Secondly, these methods require strong domain expertise or access to generative tools such as Large Language Models (LLMs) to generate descriptions of the concepts of spurious features or substantial exemplars of such features. However, the responses from generative tools are not reliable. Zhang et al. (2023b); Xu et al. (2024) indicate the existence of hallucinations in LLMs. This unreliability substantially diminishes the effectiveness of methods designed to mitigate spurious correlations through text-based modalities. Moreover, An et al. (2024); Adila et al. (2024) observe performance disparities when using different LLMs.

A recent study, ROBOSHOT (Adila et al., 2024), has been proposed to address spurious correlation issues by considering both image and text modalities. ROBOSHOT employs LLMs to generate sufficient insights for spurious features and then applies a linear projection to map image embeddings onto a neutralization hyperplane for these spurious features. This approach presents several challenges. First, the spurious insights generated by LLMs are inherently less reliable. Second, the projection operation distorts the distribution of image embeddings and significantly reduces their diversity. Third, this method lacks theoretical analysis of the optimality of the projection direction, a factor that critically influences the performance of group robustness.

To sum up, existing methods can be categorized into three types, each with specific concerns. First, some methods require target labels, violating the zero-shot classification requirements. Second, methods that focus solely on one modality face risks of misalignment when integrating different modalities. Third, approaches using linear projection distort the distribution of image embeddings. Additionally, reliance on LLMs introduces concerns regarding reliability.

To robustify zero-shot VLMs effectively, the main requirements are **no training**, **no label requirement**, **no reliance on LLMs**. To address these challenges, we propose a novel approach `TIE`, a framework that utilizes text prompt guidance to reduce spurious features in image embeddings. Contrary to the linear transformation techniques introduced in (Adila et al., 2024; Chuang et al., 2023), we adopted a *translation* operation in the latent space, which preserves the distribution of image embeddings. Our method is grounded in theoretical analysis that identifies the optimal parameter for translating image embeddings. Unlike methods that focus on a single modality, we incorporate text prompts to guide the translation operation in the image space, thereby preserving alignment across both modalities.

In practice, when spurious labels are inaccessible, we develop `TIE*`. `TIE*` leverages a zero-shot manner to infer spurious features and utilizes pseudo-spurious labels to enhance the group robustness of VLMs, without relying on manual annotations. Throughout this process, our method does not require training any parameters in VLMs, thus enhancing efficiency.

We conducted extensive experiments on real-world datasets, including high-stakes biomedical settings. The results show that our method significantly outperforms existing approaches. Additionally,

we provide visualizations to demonstrate that the proposed method effectively mitigates spurious correlations.

We summarize our contribution as follows:

- We propose a theoretically inspired method that is simple and effective in mitigating spurious correlation issues in VLMs for zero-shot classification.
- The proposed algorithm operates without the need for LLMs or labeled data, and does not require access to the internal parameters of VLMs.
- We empirically validate the effectiveness of the proposed method, including visualizations across both image and text modalities.

## 2 RELATED WORKS

### 2.1 GROUP ROBUSTNESS

Many methods have been proposed to enhance group robustness and address issues of spurious correlations (Sagawa et al., 2019; Arjovsky et al., 2019; Idrissi et al., 2022; Kirichenko et al., 2022; Liu et al., 2021; Yao et al., 2022; Krueger et al., 2021; Lu et al., 2024). These approaches predominantly utilize reweighting techniques to adjust the weights of samples in the training set. These methods are designed for single-modality classification and involve training either all or a subset of the model's parameters. In contrast, our approach significantly differs from these conventional methods as it requires no adjustments to the parameters in the backbone during the robustification process.

### 2.2 MITIGATING SPURIOUS CORRELATION IN VLMS

To mitigate spurious correlations in VLMs, many approaches focus on fine-tuning using labeled datasets. Specifically, Goyal et al. (2023) employ target labels derived from text descriptions and fine-tunes the model using a contrastive loss. Yang et al. (2023) propose a method that detects spurious attributes and fine-tunes VLMs using contrastive loss both within and across different modalities. Petryk et al. (2022) propose a framework that uses VLMs to integrate textual information with images and generate a saliency map. This map is then used to supervise the training of a classifier. Zhang & Ré (2022) propose an adapter that connects to the embedding layer and utilizes contrastive loss to fine-tune the adapter. Dehdashtian et al. (2024) propose a method that employs the Hilbert-Schmidt Independence Criterion (HSIC) to debias both image and text embeddings. Pang et al. (2024) introduce a method for distributional robustness via language that maximizes the entropy of predictions on spurious attributes. Distinct from the existing methods mentioned above, our method operates without any labeled data, thus fulfilling the requirements for zero-shot classification.

### 2.3 GROUP ROBUSTNESS IN ZERO-SHOT CLASSIFICATION

Another line of research addresses spurious correlation issues in VLMs in a zero-shot manner. Trager et al. (2023) propose a method that combines a target prompt with spurious prompts and averages them to generate an 'Ideal words' prompt. An et al. (2024) employs a two-step inference method that first identifies spurious features and then augments the text prompt with these identified features. Chuang et al. (2023) propose a method that projects text embeddings onto a space orthogonal to the spurious attribute space. Ge et al. (2023) aim to enhance text prompt robustness by focusing on label augmentation. Adila et al. (2024) propose a method that uses the Gram-Schmidt process to project representations onto a space orthogonal to spurious features. In contrast, our method does not depend on augmenting the prompt, which simplifies usage and reduces concerns about the hallucination problem in LLMs. Additionally, our approach aims to mitigate spurious correlations from a multimodal perspective.

## 3 METHODS

### 3.1 PRELIMINARIES

**Setting.** This work focuses on the group robustness setting (Sagawa et al., 2019) in the zero-shot classification task. Denote $\mathbf{x} \in \mathcal{X}$ as the input image, $y \in \mathcal{Y}$ as the target label, and $a \in \mathcal{A}$ as the spurious feature. Define group $g_{y,a} \in \mathcal{G}$ considering the combination of target label $y$ and spurious feature $a$. To mitigate the impact of spurious correlations on prediction, our approach

follows the established practices (Sagawa et al., 2019; Liu et al., 2021; Kirichenko et al., 2022) aimed at enhancing the accuracy of the worst groups while preserving overall accuracy.

**Relationship between vanilla classification and zero-shot classification.** We first bridge these two tasks for the subsequent theoretical discussion. Denote $\phi_I(\cdot)$ as the image encoder, $\phi_T(\cdot)$ as the text encoder, $\mathbf{t}_y \in \mathcal{T}$ as the text prompt, with each text prompt corresponding to one target label $y$. For example, in waterbirds dataset (Sagawa et al., 2019), for $y$ = Waterbird, $\mathbf{t}_y$ = "a photo of a waterbird", $\mathcal{T}$ = {a photo of a waterbird, a photo of a landbird }, where $|\mathcal{T}| = K$, corresponding to $K$ classes of text prompts. For zero-shot classification, the VLMs model serves as a score function that maps $\mathcal{X} \times \mathcal{T} \rightarrow \mathbb{R}$:

$$\hat{y} = \arg\max_{k \in [K]} \langle \phi_I(\mathbf{x}), \phi_T(\mathbf{t}_k) \rangle. \tag{1}$$

Equation 1 shows the zero-shot paradigm that predicts the class $\hat{y}$ as the one with the highest inner product between the image embedding and the text prompt embedding.

Vanilla classification: Denote $\mathbf{h} \in \mathbb{R}^d$ as the representation learned from a neural network, which is processed by an image encoder $\phi_I(\cdot)$, i.e. $\mathbf{h} = \phi_I(\mathbf{x})$. $\mathbf{W} = [\mathbf{w}_1, ...\mathbf{w}_k] \in \mathbb{R}^{d \times K}$ as a linear classifier. The vanilla classification task:

$$\hat{y} = \arg\max_{k \in [K]} \mathbf{W}^\top \mathbf{h} = \arg\max_{k \in [K]} \langle \phi_I(\mathbf{x}), \mathbf{W} \rangle. \tag{2}$$

Comparing Equation 2 with Equation 1, it can be concluded that the zero-shot classification represents a specialized form of vanilla classification, where the linear classifier is composed of text embeddings. For simplicity in the following analysis, we use $\mathbf{h}$ to denote $\phi_I(\mathbf{x})$ and $\mathbf{w}$ to represent $\phi_T(\mathbf{t}_y)$, based on their equivalence.

## 3.2 Theoretical analysis

**Spurious correlation modeling.** We adopt a common setting in modeling spurious correlation (Sagawa et al., 2020; Idrissi et al., 2022; Yao et al., 2022; Wang & Wang, 2024). Concretely, denote a spurious feature $a \in \{-1, 1\}$ and a label $y \in \{-1, 1\}$. Each $(y, a)$ group denoted as $g_{y,a}$ has its own distribution over the image embedding $\mathbf{h} = [h_a, h_{\text{core}}, h_{\text{noise}}] \in \mathbb{R}^d$, where

$$h_a | a \sim \mathcal{N}(a, \sigma_a^2), \ \ h_{\text{core}} | y \sim \mathcal{N}(y, \sigma_{core}^2), \ \ h_{\text{noise}} \sim \mathcal{N}(0, I). \tag{3}$$

The data model assumption is for the simplicity of the following analysis. Without loss of generality, the dimensions of core features and spurious features can be arbitrary. We investigate the problem of improving the group robustness of VLMs in a zero-shot setting by adjusting $\mathbf{h}$ given fixed target text prompts. By modeling each group with equal weight, the goal is to maximize each group-wise utility:

$$\mathcal{L}_{Acc}(\mathbf{h}_{g_{y,a}}, \mathbf{w}) = \max_{\mathbf{h}} \sum_{g_{y,a} \in \mathcal{G}} A(\mathbf{h}_{g_{y,a}}, \mathbf{w}; y), \tag{4}$$

where $A(\cdot)$ is the accuracy function, $\mathbf{h}_{g_{y,a}}$ corresponds to the image embeddings from group $g_{y,a}$. We introduce Lemma 1, which establishes that the accuracy for each group can be derived in an analytical form.

**Lemma 1** *Under the above data model assumption, the group-wise accuracy can be derived as*

$$A(\mathbf{h}_{g_{y,a}}, \mathbf{w}; y) = \begin{cases} \dfrac{1}{2} erfc(-\dfrac{\mathbf{w}^\top \boldsymbol{\mu}_{g_{y,a}}}{\sqrt{2\mathbf{w}^\top \Sigma_{g_{y,a}} \mathbf{w}}}), \ if \ y = 1 \\ \\ \dfrac{1}{2} erf(-\dfrac{\mathbf{w}^\top \boldsymbol{\mu}_{g_{y,a}}}{\sqrt{2\mathbf{w}^\top \Sigma_{g_{y,a}} \mathbf{w}}}) + \dfrac{1}{2}, \ if \ y = -1, \end{cases} \tag{5}$$

*where $\boldsymbol{\mu}_{g_{y,a}}$ and $\Sigma_{g_{y,a}}$ represent the mean and covariance matrix of the image embedding $\mathbf{h}_{g_{y,a}}$.*

The proof is presented in Appendix A. This lemma quantifies the accuracy of each $(y, a)$ group given a fixed classifier $\mathbf{w}$. According to Lemma 1, adjusting either $\boldsymbol{\mu}$ or $\Sigma$ impacts the group-wise accuracy. The solution proposed by (Adila et al., 2024) involves changing $\Sigma$, which changes the

distribution of the image embeddings in the latent space. This change necessitates a highly precise decision boundary for spurious features, as the accuracy of the worst-performing group is extremely sensitive to the accuracy of this boundary. If the boundary is not accurately defined, the worst-performing group's accuracy will significantly deteriorate. We discuss this phenomenon further and provide a theoretical comparison along with experimental validation of our approach in Section 3.3 and Appendix C.1.

**Objective.** We propose a translation operator that preserves the distribution of image embeddings. In particular, our objective function is to find the optimal translation vectors $\mathbf{v}_a$ to maximize the following objective function:

$$\mathcal{L}_{Acc}(\mathbf{v}_a; \mathbf{h}_{g_{y,a}}, \mathbf{w}) = \max_{\mathbf{v}_a} \sum_{g_{y,a} \in \mathcal{G}} A(\mathbf{h}_{g_{y,a}} + \mathbf{v}_a, \mathbf{w}; y), \tag{6}$$

$\mathbf{v}_a$ is the translation vectors based on the label of spurious features. In Theorem 1, we establish the optimal vector for translation within the complete set of feasible directions. We leave the detailed proof in Appendix B.

**Theorem 1** *Given the objective function and the data model, the maximizer of the objective is obtained by*

$$\mathbf{v}_a^* = \mathbb{E}[-\mathbf{P}\mathbf{h}_a], \tag{7}$$

*where $\mathbf{P} \in \mathbb{R}^{d \times d}$ is an elementary matrix,* $\mathbf{P} = \begin{pmatrix} 1 & 0 & \cdots & 0 \\ 0 & 0 & \cdots & 0 \\ \vdots & \vdots & \ddots & \vdots \\ 0 & 0 & \cdots & 0 \end{pmatrix}$.

Theorem 1 states that the optimal translate vector $\mathbf{v}_a$ can be computed by $\mathbf{v}_a = \mathbb{E}[-h_{\text{spu}}, 0, ..., 0]$, which is *the negative direction of the spurious feature vector*. However, estimating the spurious feature vector presents a challenge. Wu et al. (2023) proposed first training a classifier to classify the spurious feature and then using the vector orthogonal to the decision hyperplane as the spurious feature vector. We argue that this method significantly compromises efficiency as the need for training and risks misalignment in the text embedding space. In the realm of VLMs, effectively combining both text and image embeddings is crucial. Therefore, we propose using spurious text embeddings to guide image embeddings toward an optimal state.

### 3.3 TIE: Text prompt based Image embedding translation

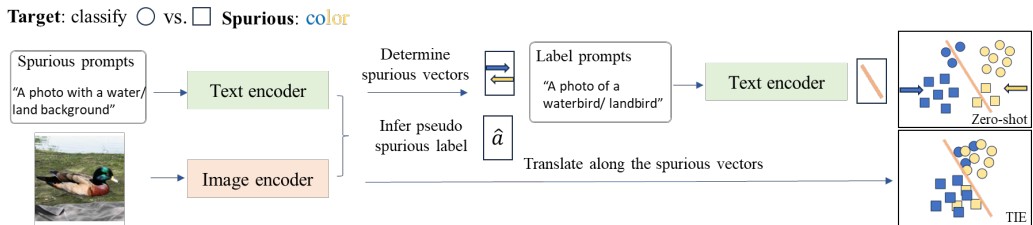

Figure 2: TIE⋆ overview. First, we utilize spurious prompts to compute the spurious vectors. We then employ the CLIP model to infer the spurious label for each sample. Subsequently, we translate the image embeddings along the spurious vector based on the pseudo-spurious label. Finally, we use these translated embeddings to perform the zero-shot classification task.

We now present our method to mitigate spurious correlations in the VLMs, an overview is shown in Figure 2. Based on the analysis in Section 3.2, we first compute the spurious feature vector. Next, we translate the image embeddings along the opposite of this direction, and then use the adjusted image embeddings to perform zero-shot classification.

**Computation on Spurious feature vector.** Given a set of spurious text prompts $\mathcal{T}_a$ (e.g. `a photo with a water background, a photo with a land background`). TIE computes the spurious vector $\mathbf{v}_a = \phi_T(\mathbf{t}_a; a)$, *s.t.* $\mathbf{t}_a \in \mathcal{T}_a$. TIE normalizes $\mathbf{v}_a$ by its $\mathcal{L}_2$ norm: $\mathbf{v}_a = \frac{\mathbf{v}_a}{||\mathbf{v}_a||_2}$.

**Translate the image embeddings.** Given an image, `TIE` first computes its image embedding using the image encoder, i.e., $\mathbf{h}_a = \phi_I(\mathbf{x}; a)$. Then, `TIE` computes the magnitude of the translation by $\lambda_a = \mathbb{E}[\mathbf{h}_a^\top \mathbf{v}_a]$, which is the average projection length on the direction of $\mathbf{v}_a$. Next, `TIE` translates image embedding by

$$\mathbf{h}_a \leftarrow \mathbf{h}_a - \lambda_a \mathbf{v}_a. \tag{8}$$

The zero-shot classification task employs $\mathbf{h}_a$ and target prompts for execution.

**Without spurious feature label.** One constraint on `TIE` is its dependency on access to labels for spurious features, with samples bearing various spurious labels moving in different directions to achieve an optimal state. To address this, we propose `TIE*` that eliminates the need for any labeled data within the dataset.

An et al. (2024) empirically demonstrated that spurious features can be effectively inferred using VLMs. Building upon this insight, we leverage VLMs to infer the spurious labels for each sample in the dataset. Concretely, we assign a Pseudo-spurious label in the zero-shot classification setting:

$$\hat{a} = \arg\max_{a \in \mathcal{A}} \langle \phi_I(\mathbf{x}), \phi_T(\mathbf{t}_a) \rangle \tag{9}$$

where $\hat{a}$ is the pseudo-spurious label for the sample. In equation 9, the pseudo-labeling procedure requires of all possible spurious text prompts. We utilize these pseudo-labeled to implement the corresponding translation operation as introduced in the previous section. We summarize our method in Algorithm 1.

We conduct experiments under two scenarios: In the first, where the labeled spurious feature is available, we apply the true spurious label to implement `TIE`. In the second scenario, where the labeled spurious feature is unavailable, we execute the complete algorithm as outlined in Algorithm 1, denoted as `TIE*`. Additionally, we investigate a method applicable when partially labeled data is available. The detailed discussion of this method is deferred to Section 4.4.

### 3.4 Theoretical comparison between `TIE` and `ROBOSHOT`

`TIE` and `ROBOSHOT` are methods designed to address spurious correlations by leveraging both image and text modalities. We provide a detailed comparison of the worst group accuracy between two methods under different spurious text prompts and label prompts. To quantify the effects of spurious text prompts and target label text prompts, as discussed in 3.1, these prompts form two classifiers: $\mathbf{w}_a$ for spurious prompts and $\mathbf{w}$ for label prompts. We define $\mathbf{w}_a = [1, \alpha, \mathbf{0}]$ and $\mathbf{w} = [1, \beta, \mathbf{0}]$, $\alpha, \beta \in \mathbb{R}^+$ A smaller $\alpha$ indicates more accurate spurious decision boundary, while a larger $\beta$ indicates a more accurate task boundary. Utilizing these definitions, we have the analytical forms for the worst group accuracy (WG) for both ROBOSHOT and TIE:

$$\text{ROBOSHOT}: WG_{RS}(\alpha, \beta) = \min\{\frac{1}{2}\text{erfc}(-\frac{\alpha^2 - (1+\beta)\alpha + \beta}{\sqrt{2}(1+\alpha^2)(1+\alpha\beta)}),$$
$$\frac{1}{2}\text{erf}(-\frac{\alpha^2 - (\beta-1)\alpha - \beta}{\sqrt{2}(1+\alpha^2)(1+\alpha\beta)}) + \frac{1}{2}\}. \tag{10}$$

$$\text{TIE}: WG_{TIE}(\alpha, \beta) = \min\{\frac{1}{2}\text{erfc}(-\frac{(1+\beta)\sqrt{1+\alpha^2} - \alpha\beta - 1}{\sqrt{2}(1+\beta^2)(1+\alpha^2)}),$$
$$\frac{1}{2}\text{erf}(-\frac{(1-\beta)\sqrt{1+\alpha^2} + \alpha\beta - 1}{\sqrt{2}(1+\beta^2)(1+\alpha^2)}) + \frac{1}{2}\}. \tag{11}$$

We defer the derivation of equations 10 and 11 in Appendix C. We present a plot of the theoretical worst group accuracy with respect to $\alpha$ and $\beta$ in Figure 3. We observe that `ROBOSHOT` only achieves a higher WG when $\alpha \to 0$, representing the perfect spurious classifier. Otherwise, `ROBOSHOT`'s performance drops rapidly when the spurious classifier is inaccurately approximated, showing a significant margin compared to `TIE`. In other words, the performance of `TIE` shows better robustness across different text prompts. We further substantiate this analysis with empirical validation on a real-world dataset, as detailed in Appendix C.1.

## 4 Experiment

### 4.1 Setup

**Datasets.** We study five well-established benchmark datasets for spurious correlation research: Waterbirds (Koh et al., 2021; Sagawa et al., 2019), CelebA (Liu et al., 2015), ISIC (Codella et al., 2019), COVID-19 (Cohen et al., 2020), FMOW (Christie et al., 2018). Please refer to appendix E for detailed information.

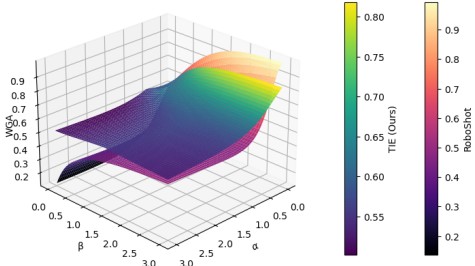

Figure 3: Theoretical comparison of worst group accuracy between TIE and ROBOSHOT.

**Backbones.** Existing research indicates that different visual backbones produce varied results. Following established protocols (Adila et al., 2024), for the Waterbirds and ISIC datasets, we examine CLIP models with vision backbone of ViT-B/32, ViT-L/14, and RN50 (Ilharco et al., 2021; Cherti et al., 2023; Radford et al., 2021). For the ISIC and COVID-19 datasets, we utilize Biomed CLIP (Zhang et al., 2023a) as the vision backbone. For the FMoW dataset, we employ the ViT-L/14 model due to the dataset's complex nature.

**Baselines.** We compare our method against two baselines and existing state-of-the-art methods in robust zero-shot classification. Concretely, two baselines are vanilla zero-shot classification (ZS), Zero-shot with group information (Group prompt). Existing SOTA methods including Ideal Prompt (Trager et al., 2023), Orth-Cali (Chuang et al., 2023), Perception CLIP (An et al., 2024), RO-BOSHOT (Adila et al., 2024). We leave the details of baselines in Appendix F.

**Text Prompts for Reproducibility.** Zero-shot classification employs two types of text prompts: label prompts and spurious prompts. To ensure a fair comparison, all methods utilize the same label prompts. For example, the label prompts for the Waterbirds dataset are [a photo of a landbird, a photo of a waterbird]. For spurious prompts, we use the prompts provided by the authors if the method is tested on a specific dataset. Otherwise, we generate spurious prompts using generative AI tools like ChatGPT (OpenAI, 2023), following the guidelines specified in the original papers. For reproducibility [1], prompts used in our experiments are provided in Appendix G.

**Metrics.** Following the protocol established by robust learning studies (Sagawa et al., 2019; Adila et al., 2024), we report three metrics: worst group accuracy (WG), average accuracy (Avg), and the gap between these two metrics (Gap). We highlight the best result in **bold** and underline the second-best result.

## 4.2 MAIN RESULTS.

**Waterbirds.** Table 1 summarizes results on the Waterbirds dataset. TIE achieves significant improvement over comparative methods by a relatively large margin, especially for the ViT-L14 vision backbone, where the worst group accuracy reaches 78.82%, surpassing the previous method by 14.65%. TIE⋆ achieves a comparable performance in the ViT backbones. However, performance varies with different backbone models. For ResNet-50, Orth-Cali outperforms other methods.

Table 1: Zero Shot classification results on Waterbirds

| Method | CLIP (ViT-B32) | | | CLIP (ViT-L14) | | | CLIP (ResNet-50) | | |
|---|---|---|---|---|---|---|---|---|---|
| | WG ↑ | Avg ↑ | Gap ↓ | WG ↑ | Avg ↑ | Gap ↓ | WG ↑ | Avg ↑ | Gap ↓ |
| ZS | 41.37 | 68.48 | 27.11 | 31.93 | 83.72 | 51.79 | 35.36 | 80.64 | 45.28 |
| Group Prompt | 43.46 | 66.79 | 23.33 | 10.44 | 56.12 | 45.68 | 49.84 | 70.96 | 21.12 |
| Ideal words | 60.28 | 79.20 | 18.92 | 64.17 | **87.67** | 23.50 | 39.09 | 79.48 | 40.39 |
| Orth-Cali | 54.99 | 69.19 | 14.20 | 58.56 | 86.31 | 27.75 | **64.80** | 84.47 | **19.67** |
| Perception CLIP | 59.78 | **82.50** | 22.72 | 54.12 | 86.74 | 32.62 | 48.21 | **91.51** | 43.30 |
| ROBOSHOT | 54.41 | 71.92 | 17.51 | 45.17 | 64.43 | 19.26 | 26.61 | 69.06 | 42.45 |
| TIE (Ours) | **71.35** | 79.82 | **8.47** | **78.82** | 84.12 | **5.30** | 52.96 | 83.62 | 30.66 |
| TIE⋆ (Ours*) | 61.24 | 76.91 | 15.67 | 61.60 | 78.98 | 17.38 | 34.11 | 81.19 | 47.08 |

---

[1] Code at https://github.com/lu876/TIE

**CelebA.** Table 2 presents results for the CelebA dataset. Similar to the Waterbirds dataset, `TIE` consistently outperforms comparison baselines and achieves the smallest gap in ViT backbone models. The performance of `TIE*` is comparable to that of `TIE`. For the ResNet backbone, Perception CLIP outperforms other methods.

Table 2: Zero Shot classification results on CelebA

| Method | CLIP (ViT-B32) | | | CLIP (ViT-L14) | | | CLIP (ResNet-50) | | |
|---|---|---|---|---|---|---|---|---|---|
| | WG ↑ | Avg ↑ | Gap ↓ | WG ↑ | Avg ↑ | Gap ↓ | WG ↑ | Avg ↑ | Gap ↓ |
| ZS | 78.89 | 84.27 | 5.38 | 73.35 | 81.20 | 7.85 | 69.69 | 81.58 | 11.89 |
| Group Prompt | 74.90 | 80.38 | 5.48 | 68.94 | 77.86 | 8.92 | 70.59 | 79.48 | 8.89 |
| Ideal words | 78.12 | 80.96 | 2.84 | 76.67 | **89.15** | 12.48 | 65.65 | 76.27 | 10.62 |
| Orth-Cali | 77.92 | 82.31 | 4.39 | 77.69 | 81.39 | 3.70 | 69.13 | 76.47 | 7.34 |
| Perception CLIP | 76.46 | 80.32 | 3.86 | 78.70 | 81.41 | 2.71 | **80.22** | **85.17** | **4.95** |
| ROBOSHOT | 80.52 | 84.77 | 4.25 | 82.61 | 85.54 | 2.93 | 73.96 | 80.90 | 6.94 |
| `TIE` (Ours) | **82.63** | **85.11** | **2.48** | **84.60** | 86.17 | **1.57** | 75.32 | 81.71 | 6.39 |
| `TIE*` (Ours*) | 82.61 | 85.10 | 2.49 | 81.98 | 84.27 | 2.29 | 75.30 | 81.70 | 6.40 |

**ISIC and COVID-19.** Our experiments extend to specialty datasets within high-stakes settings, specifically deploying VLM models in the medical domain. Table 3 shows the results for the ISIC and COVID-19 datasets where our method outperforms baseline methods in worst-group accuracy and achieves comparable average accuracy.

Table 3: Zero Shot classification results on ISIC and Covid-19 datasets

| Method | ISIC (Biomed CLIP) | | | COVID-19 (Biomed CLIP) | | |
|---|---|---|---|---|---|---|
| | WG ↑ | Avg ↑ | Gap ↓ | WG ↑ | Avg ↑ | Gap ↓ |
| ZS | 42.21 | 70.21 | 28.00 | 44.83 | 61.81 | 16.98 |
| Group Prompt | 12.13 | 30.05 | 17.92 | 27.58 | 48.27 | 20.69 |
| Ideal words | 41.42 | 53.07 | 11.65 | 23.53 | 56.84 | 33.31 |
| Orth-Cali | 21.43 | **72.54** | 51.11 | 44.83 | 51.72 | **6.89** |
| Perception CLIP | 41.55 | 52.74 | 11.19 | 48.84 | 56.87 | 8.03 |
| ROBOSHOT | 53.30 | 59.84 | 6.54 | 32.75 | 53.10 | 20.35 |
| `TIE` (Ours) | **65.87** | 69.90 | **4.03** | **52.17** | **62.50** | 10.33 |
| `TIE*` (Ours*) | 61.11 | 71.68 | 10.57 | 50.22 | 61.08 | 10.86 |

**FMOW.** We extend our experiments to multiclasses and multigroup settings. The FMOW dataset includes 62 classes and is organized into 5 spurious groups. Table 4 shows the results for FMOW. `TIE` achieves the highest accuracy in the worst-performing group, `TIE*` shows comparable performance on the worst group accuracy and has the highest overall accuracy. These results further validate the effectiveness of our methods in mitigating spurious correlations in the zero-shot setting.

Table 4: Top-1 Accuracy and Worst Group accuracy on FMOW dataset.

| | WG ↑ | Avg ↑ | Gap ↓ |
|---|---|---|---|
| ZS | 18.06 | 26.02 | 7.96 |
| Group Prompt | 8.75 | 14.69 | 5.94 |
| Ideal words | 11.14 | 20.21 | 9.07 |
| Orth-Cali | 19.45 | 26.11 | 6.66 |
| Perception CLIP | 12.61 | 17.70 | **5.09** |
| ROBOSHOT | 10.88 | 19.79 | 8.91 |
| `TIE` | **20.19** | 26.62 | 6.43 |
| `TIE*` | 19.84 | **26.65** | 6.81 |

**Discussion.** From Table 1-4, `TIE` consistently achieves the best or second-best WG, `TIE*` achieves a comparable result but still has a performance gap, which will be discussed in the following section. We found `TIE` shows relative suboptimal performance using ResNet-50 on the Waterbirds

dataset. Note that all text encoders are transformer-based models, while the vision backbones vary. We hypothesize that this suboptimality primarily arises from a misalignment between the direction of the spurious vector in the text space and the image space. This misalignment stems from the structure and scales of the encoders, which echoes the finding that different CLIP structures show significantly different zero-shot classification results (Radford et al., 2021). Methods like Orth-Cali or Perception CLIP, which only focus on debiasing text embeddings, introduce randomness into zero-shot classification. This randomness can occasionally enhance performance. However, adjusting text embeddings without considering image embeddings can result in misalignment, leading to a significant drop in performance. For example, Orth-Cali shows suboptimal performance on the ISIC dataset. Conversely, our method mitigates this randomness by integrating both image and text modalities, thereby enhancing the stability of zero-shot classification outcomes.

## 4.3 Group robust text prompt

In this section, we demonstrate that our method is compatible with other methods focused on mitigating spurious correlations in the text modality. An et al. (2024) highlight that providing additional context enhances the performance of VLM models. Inspired by this insight, we employed group-robust prompts to identify spurious directions. Specifically, we utilize GPT-4 (OpenAI, 2023) to generate five sentences that serve as synonyms for spurious features. The prompt for the GPT-4 is `Please generate 5 synonyms of [Spurious feature]`. For instance, the robustified spurious prompts for the Waterbirds dataset include: for a land background, [`A photo with a land background. A photo of a forest background. A photo of a mountain background. A photo of a Terrain background. A photo of a Ground background`]; and for a water background, [`A photo with a water background. A photo of an ocean background. A photo of a sea background. A photo of a Lake background. A photo of a River background.`]. We computed the average text embedding from these spurious prompts and used it to update the image embedding. The results are shown in Table 5. We observe that the robustified prompt helps find a more robust direction for the spurious features, leading to improved WG and Avg metrics with ViT-B32 and ResNet-50 models.

Table 5: Group robustify prompting

| Method | ViT-B-32 | | | ViT-L-14 | | | ResNet-50 | | |
|---|---|---|---|---|---|---|---|---|---|
| | WG ↑ | Avg ↑ | Gap ↓ | WG ↑ | Avg ↑ | Gap ↓ | WG ↑ | Avg ↑ | Gap ↓ |
| TIE⋆ | 61.24 | 76.91 | 15.67 | **61.60** | **78.98** | 17.38 | 34.11 | 81.19 | 47.08 |
| TIE⋆ Robust | **64.96** | **78.63** | **13.67** | 61.46 | 78.46 | **17.00** | **38.63** | **82.22** | **43.59** |

## 4.4 Limited access to labels of the spurious features

Table 1 reveals a performance disparity between `TIE` and `TIE⋆`, suggesting that accurate estimation of the spurious label enhances performance. Wang & Wang (2024) theoretically demonstrates that feature separability directly influences performance, especially when spurious features are more separable than core features. Based on this, accurately predicting labels of the spurious features necessitates significantly fewer training samples. Therefore, we propose using a partially spurious feature labeled dataset to infer the spurious labels of the entire dataset, and subsequently apply our algorithm based on the pseudo labels of the spurious feature. We tested this approach on the Waterbirds dataset with training sample sizes ranging from 100 to 1000. To optimize efficiency, we employed a smaller-scale architecture, ResNet-18 (He et al., 2016), to predict the pseudo-spurious feature labels. The model was trained using an SGD optimizer with a learning rate of $10^{-4}$, a weight decay of $10^{-3}$, and a momentum of 0.9, over 200 epochs. The VLM model is tested using ViTB-32.

Figure 4 reports the outcomes utilizing different sample sizes within the training set. Observations indicate that increasing the amount of labeled data enhances the worst group accuracy of the CLIP model. Specifically, using 1000 samples, performance nearly matches that of our method when attribute $a$ is known. Additionally, the figure demonstrates a nearly linear improvement in worst group accuracy as the accuracy of predictions on spurious feature labels increases in the CLIP model.

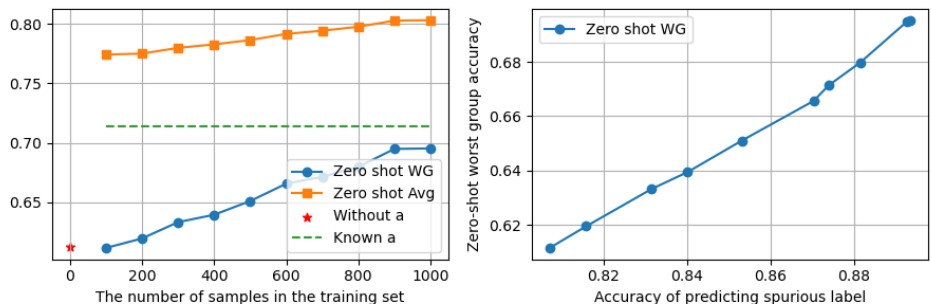

Figure 4: Performance on the Waterbirds dataset using partially labeled spurious features.

### 4.5 VISUALIZATION

In addition to the superior performance of our method, we further investigate its capacity to ensure that predictions are correct for the right reasons. This can be verified through visual explanation maps, as illustrated in Figure 5. We employed the explainability method from (Chefer et al., 2021) to generate heatmaps for both image features and text prompts. Our method significantly reduces reliance on spurious features in a zero-shot setting. In the ISIC dataset, it specifically minimizes attention to irrelevant color patches. For samples of malignant lesions, our approach enhances focus on the lesion itself rather than the other skin part. For the Waterbirds dataset, even in the vanilla zero-shot where the focus might incorrectly shift to the background, our method effectively redirects attention towards the core features of the subject. Interestingly, after implementing our method, the text prompts also show increased attention to specific objects, such as `bird` and `malignant`.

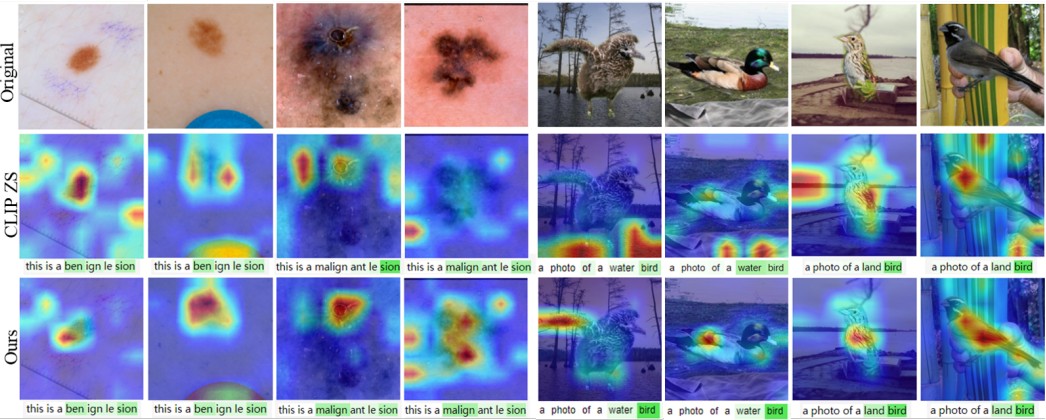

Figure 5: Attention based explanations (Chefer et al., 2021) for ISIC and Waterbirds datasets.

## 5 CONCLUSION

Addressing spurious correlations presents a critical challenge in the realm of zero-shot VLMs. This study draws inspiration from rigorous theoretical analysis to examine optimal strategies for translating image embeddings. To address the spurious correlations effectively, we have designed the `TIE` algorithm, which guides the translation of image embeddings based on the text prompt. Extensive experiments conducted on real-world datasets demonstrate that our method not only significantly improves the worst-group accuracy across all datasets but also achieves comparable overall accuracy. Additionally, we visualize results from both modalities to confirm that the predictions are based on valid reasons.

**Failure case discussion and Future direction.** Although our proposed method demonstrates significant robustness, `TIE*` may encounter failures when pseudo-spurious labels are incorrectly assigned. We present a comprehensive analysis of these failure cases and propose solutions in Appendix K. Additionally, `TIE` faces limitations when processing images with artifacts. We discuss these issues in detail in Appendix J. Identifying such artifacts could be a promising direction for future research to enhance zero-shot classification performance.

ACKNOWLEDGEMENTS

This work was partially supported by the EMBRIO Institute, contract #2120200, a National Science Foundation (NSF) Biology Integration Institute, and NSF IIS #1955890, IIS #2146091, IIS #2345235.

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

## A    PROOF OF LEMMA 1

**Lemma 1** *Under the above data model, the group-wise accuracy can be derived as*

$$A(\mathbf{h}_{g_{y,a}}, \mathbf{w}; y) = \begin{cases} \dfrac{1}{2}\mathrm{erfc}(-\dfrac{\mathbf{w}^\top \boldsymbol{\mu}_{g_{y,a}}}{\sqrt{2\mathbf{w}^\top \Sigma_{g_{y,a}} \mathbf{w}}}), \text{ if } y = 1 \\[4mm] \dfrac{1}{2}\mathrm{erf}(-\dfrac{\mathbf{w}^\top \boldsymbol{\mu}_{g_{y,a}}}{\sqrt{2\mathbf{w}^\top \Sigma_{g_{y,a}} \mathbf{w}}}) + \dfrac{1}{2}, \text{ if } y = -1 \end{cases} \tag{12}$$

*where $\boldsymbol{\mu}_{g_{y,a}}$ and $\Sigma_{g_{y,a}}$ represent the mean and covariance matrix of the image embedding $\mathbf{h}_{g_{y,a}}$.*

Denote the linear classifier as $\mathbf{w} \in \mathbb{R}^d$. To simplify the notation, we drop the subscript of $g_{y,a}$. The hyperplane is defined as two half-spaces:

$$\begin{aligned} \Omega_+ &= \{\mathbf{h}|\boldsymbol{w}^\top \mathbf{h} > 0\}, \\ \Omega_- &= \{\mathbf{h}|\boldsymbol{w}^\top \mathbf{h} \leq 0\}. \end{aligned} \tag{13}$$

The probability density function can be written as:

$$f_{\mathbf{H}}(\mathbf{h}; \boldsymbol{\mu}, \Sigma) = \frac{1}{(2\pi)^{d/2}\sqrt{\det\Sigma}} \exp(-\frac{1}{2}(\mathbf{h} - \boldsymbol{\mu})^\top \Sigma^{-1}(\mathbf{h} - \boldsymbol{\mu})). \tag{14}$$

We first consider $y = 1$. For computing the group accuracy, we integrate $f_{\mathbf{H}}(\mathbf{h}; \boldsymbol{\mu}, \Sigma)$ over the region of $\Omega_+$. In the following proof, we omit the input of $A(\cdot)$ for simplicity:

$$A = \int_{\Omega_+} f_{\mathbf{H}}(\mathbf{h}; \boldsymbol{\mu}, \Sigma)d\mathbf{h}. \tag{15}$$

Transform $\mathbf{h}$ to reduce the mean term, define $\mathbf{h}' = \mathbf{h} - \boldsymbol{\mu}$, $\Omega_1 = \{\mathbf{h}'|\boldsymbol{w}^\top \mathbf{h}' + \boldsymbol{w}^\top \boldsymbol{\mu} > 0\}$ Fischer (2013):

$$A = \frac{1}{(2\pi)^{d/2}\sqrt{\det\Sigma}} \int_{\Omega_1} \exp(-\frac{1}{2}\mathbf{h}'^\top \Sigma^{-1}\mathbf{h}')d\mathbf{h}', \tag{16}$$

$\Sigma$ is a positive definite matrix, we have $\Sigma = \mathbf{Q}^\top \Sigma' \mathbf{Q}$, where $\mathbf{Q}$ is an orthogonal matrix, and $\Sigma'$ is a diagonal matrix. We solve $\Sigma^{-1} = \mathbf{Q}^\top \Sigma'^{-1}\mathbf{Q}$.

$$A = \frac{1}{(2\pi)^{d/2}\sqrt{\det\Sigma}} \int_{\Omega_1} \exp(-\frac{1}{2}\mathbf{h}'^\top \mathbf{Q}^\top \Sigma'^{-1}\mathbf{Q}\mathbf{h}')d\mathbf{h}', \tag{17}$$

Denote $\mathbf{h}'' = \mathbf{Q}\mathbf{h}'$, $\Omega_2 = \{\mathbf{h}'' : \mathbf{w}^\top \mathbf{Q}^\top \mathbf{h}'' + \mathbf{w}^\top \boldsymbol{\mu} > 0\}$, then Equation 17 becomes

$$\begin{aligned} A &= \frac{1}{(2\pi)^{d/2}\sqrt{\det\Sigma}} \int_{\Omega_2} \exp(-\frac{1}{2}\mathbf{h}''^\top \boldsymbol{\Sigma}'^{-1}\mathbf{h}'')|\det\mathbf{Q}|d\mathbf{h}'', \\ &= \frac{1}{(2\pi)^{d/2}\sqrt{\det\Sigma}} \int_{\Omega_2} \exp(-\frac{1}{2}\mathbf{h}''^\top \Sigma'^{-1}\mathbf{h}'')d\mathbf{h}''. \end{aligned} \tag{18}$$

Eliminate the covariance term by defining $\mathbf{h}''' = \sqrt{\Sigma'^{-1}}^\top \mathbf{h}''$, $\Omega_3 = \{\mathbf{h}''' : \mathbf{w}^\top \mathbf{Q}^\top \sqrt{\Sigma'}\mathbf{h}''' + \mathbf{w}^\top \boldsymbol{\mu} > 0\}$.

Then Equation 18 becomes:

$$\begin{aligned} A &= \frac{|\sqrt{\det\Sigma'}|}{(2\pi)^{d/2}\sqrt{\det\Sigma}} \int_{\Omega_3} \exp(-\frac{1}{2}\mathbf{h}'''^\top \mathbf{h}''')d\mathbf{h}''', \\ &= \frac{1}{(2\pi)^{d/2}} \int_{\Omega_3} \exp(-\frac{1}{2}\mathbf{h}'''^\top \mathbf{h}''')d\mathbf{h}'''. \end{aligned} \tag{19}$$

The space $\Omega_3 = \{\mathbf{h}''' : \mathbf{w}'^\top \mathbf{h}''' + \mathbf{w}^\top \boldsymbol{\mu} > 0\}$, where $\mathbf{w}' = \sqrt{\Sigma'}\mathbf{Q}\mathbf{w}$.

Define an orthogonal matrix $\mathbf{U}$ s.t. $\mathbf{U}\mathbf{w}' = ||\mathbf{w}'||e$. Define $\mathbf{h}'''' = \mathbf{U}\mathbf{h}'''$, $\Omega_4 = \{\mathbf{h}'''' : ||\mathbf{w}'||e^\top\mathbf{h}'''' + \mathbf{w}^\top\boldsymbol{\mu} > 0\}$. $||\mathbf{w}'|| = \sqrt{(\sqrt{\boldsymbol{\Sigma}'}\mathbf{Q}\mathbf{w})^\top(\sqrt{\boldsymbol{\Sigma}'}\mathbf{Q}\mathbf{w})} = \sqrt{\mathbf{w}^\top\boldsymbol{\Sigma}\mathbf{w}}$. We have

$$
\begin{aligned}
A(\mathbf{h}_{g_{y,a}}, \mathbf{w}; y) &= \frac{1}{\sqrt{2\pi}} \int_{-\frac{\mathbf{w}^\top\boldsymbol{\mu}}{\sqrt{\mathbf{w}^\top\boldsymbol{\Sigma}\mathbf{w}}}}^{\infty} \exp(-\frac{1}{2}h^2)\mathrm{d}h \\
&= \frac{1}{2}\mathrm{erfc}(-\frac{\mathbf{w}^\top\boldsymbol{\mu}}{\sqrt{2}\sqrt{\mathbf{w}^\top\boldsymbol{\Sigma}\mathbf{w}}}), \text{ if } y = 1.
\end{aligned}
\tag{20}
$$

Similarly, for $y = -1$, consider integration over the region of $\Omega_-$:

$$
\begin{aligned}
A(\mathbf{h}_{g_{y,a}}, \mathbf{w}; y) &= \frac{1}{\sqrt{2\pi}} \int_{-\infty}^{-\frac{\mathbf{w}^\top\boldsymbol{\mu}}{\sqrt{\mathbf{w}^\top\boldsymbol{\Sigma}\mathbf{w}}}} \exp(-\frac{1}{2}h^2)\mathrm{d}h \\
&= \frac{1}{2}\mathrm{erf}(-\frac{\mathbf{w}^\top\boldsymbol{\mu}}{\sqrt{2}\sqrt{\mathbf{w}^\top\boldsymbol{\Sigma}\mathbf{w}}}) + \frac{1}{2}, \text{ if } y = -1.
\end{aligned}
\tag{21}
$$

Thus prove the statement $\square$.

## B    PROOF OF THEOREM 1

**Theorem 1** *Given the objective function and the data model, the maximizer of the objective is obtained by*

$$
\mathbf{v}_a = \mathbb{E}[-\mathbf{P}\mathbf{h}_a]
\tag{22}
$$

*where $\mathbf{P} \in \mathbb{R}^{d\times d}$ is an elementary matrix,* $\mathbf{P} = \begin{pmatrix} 1 & 0 & \cdots & 0 \\ 0 & 0 & \cdots & 0 \\ \vdots & \vdots & \ddots & \vdots \\ 0 & 0 & \cdots & 0 \end{pmatrix}$.

We rewrite the objective function to ensure the completeness of the proof. We first solve for the stationary point, then verify the stationary point is a local maximum point.

$$
\mathcal{L}_{Acc}(\mathbf{v}_a; \mathbf{h}_{g_{y,a}}, \mathbf{w}) = \max_{\mathbf{v}_a} \sum_{g_{y,a}\in\mathcal{G}} A_{g_{y,a}}(\mathbf{h}_{g_{y,a}} + \mathbf{v}_a, \mathbf{w}; y).
\tag{23}
$$

To maximize the objective function, the stationary point can be computed by $\nabla_{\mathbf{v}_a}\mathcal{L}_{Acc} = \mathbf{0}$:

$$
\nabla_{\mathbf{v}_a}\mathcal{L}_{Acc} = \sum_{g_{y,a}\in\mathcal{G}} \nabla_{\mathbf{v}_a} A(\mathbf{h}_{g_{y,a}} + \mathbf{v}_a) = \mathbf{0}.
\tag{24}
$$

With Lemma 1, we have

$$
\nabla_{\mathbf{v}_a}\mathcal{L}_{Acc} = \nabla_{\mathbf{v}_a} \left( \frac{1}{2}\mathrm{erfc}(-\frac{\mathbf{w}^\top(\boldsymbol{\mu}_{g_{1,a}} + \mathbf{v}_a)}{\sqrt{2\mathbf{w}^\top\boldsymbol{\Sigma}\mathbf{w}}}) + \frac{1}{2}\mathrm{erf}(-\frac{\mathbf{w}^\top(\boldsymbol{\mu}_{g_{-1,a}} + \mathbf{v}_a)}{\sqrt{2\mathbf{w}^\top\boldsymbol{\Sigma}\mathbf{w}}}) + \frac{1}{2} \right) = \mathbf{0}.
\tag{25}
$$

Decompose Equation 25 based on $a$, we first compute $\mathbf{v}_1$:

$$
\begin{aligned}
&\nabla_{\mathbf{v}_1} \left( \frac{1}{2}\mathrm{erfc}(-\frac{\mathbf{w}^\top(\boldsymbol{\mu}_{g_{1,1}} + \mathbf{v}_1)}{\sqrt{2\mathbf{w}^\top\boldsymbol{\Sigma}\mathbf{w}}}) + \frac{1}{2}\mathrm{erf}(-\frac{\mathbf{w}^\top(\boldsymbol{\mu}_{g_{-1,1}} + \mathbf{v}_1)}{\sqrt{2\mathbf{w}^\top\boldsymbol{\Sigma}\mathbf{w}}}) + \frac{1}{2} \right) \\
&= \frac{\mathbf{w}}{\sqrt{2\pi}\mathbf{w}^\top\boldsymbol{\Sigma}\mathbf{w}}[\exp(-(\frac{\mathbf{w}^\top(\boldsymbol{\mu}_{g_{1,1}} + \mathbf{v}_1)}{\sqrt{2\mathbf{w}^\top\boldsymbol{\Sigma}\mathbf{w}}})^2) - \exp(-(\frac{\mathbf{w}^\top(\boldsymbol{\mu}_{g_{-1,1}} + \mathbf{v}_1)}{\sqrt{2\mathbf{w}^\top\boldsymbol{\Sigma}\mathbf{w}}})^2)] = \mathbf{0}.
\end{aligned}
\tag{26}
$$

As $\mathbf{w} \neq \mathbf{0}$, $\mathbf{w}^\top\boldsymbol{\Sigma}\mathbf{w} \neq 0$ equation 26 can be rewritten as

$$
\exp(-(\frac{\mathbf{w}^\top(\boldsymbol{\mu}_{g_{1,1}} + \mathbf{v}_1)}{\sqrt{2\mathbf{w}^\top\boldsymbol{\Sigma}\mathbf{w}}})^2) = \exp(-(\frac{\mathbf{w}^\top(\boldsymbol{\mu}_{g_{-1,1}} + \mathbf{v}_1)}{\sqrt{2\mathbf{w}^\top\boldsymbol{\Sigma}\mathbf{w}}})^2).
\tag{27}
$$

Taking the natural log of both sides implies:

$$-(\frac{\mathbf{w}^\top(\boldsymbol{\mu}_{g_{1,1}} + \mathbf{v}_1)}{\sqrt{2\mathbf{w}^\top\boldsymbol{\Sigma}\mathbf{w}}})^2 = -(\frac{\mathbf{w}^\top(\boldsymbol{\mu}_{g_{-1,1}} + \mathbf{v}_1)}{\sqrt{2\mathbf{w}^\top\boldsymbol{\Sigma}\mathbf{w}}})^2, \tag{28}$$

or equivalently

$$\mathbf{w}^\top(\boldsymbol{\mu}_{g_{1,1}} + \mathbf{v}_1) = \pm\mathbf{w}^\top(\boldsymbol{\mu}_{g_{-1,1}} + \mathbf{v}_1). \tag{29}$$

It can be solved as:

$$\mathbf{v}_1^* = -\frac{1}{2}\sum_{y\in\{-1,1\}}\boldsymbol{\mu}_{g_{y,1}}. \tag{30}$$

Then, compute $\mathbf{v}_{-1}$,

$$\nabla_{\mathbf{v}_{-1}}\left(\frac{1}{2}\mathrm{erfc}(-\frac{\mathbf{w}^\top(\boldsymbol{\mu}_{g_{1,-1}} + \mathbf{v}_{-1})}{\sqrt{2\mathbf{w}^\top\boldsymbol{\Sigma}\mathbf{w}}}) + \frac{1}{2}\mathrm{erf}(-\frac{\mathbf{w}^\top(\boldsymbol{\mu}_{g_{-1,-1}} + \mathbf{v}_{-1})}{\sqrt{2\mathbf{w}^\top\boldsymbol{\Sigma}\mathbf{w}}} + \frac{1}{2})\right)$$

$$= \frac{\mathbf{w}}{\sqrt{2\pi}\mathbf{w}^\top\boldsymbol{\Sigma}\mathbf{w}}[\exp(-(\frac{\mathbf{w}^\top(\boldsymbol{\mu}_{g_{1,-1}} + \mathbf{v}_{-1})}{\sqrt{2\mathbf{w}^\top\boldsymbol{\Sigma}\mathbf{w}}})^2) - \exp(-(\frac{\mathbf{w}^\top(\boldsymbol{\mu}_{g_{-1,-1}} + \mathbf{v}_{-1})}{\sqrt{2\mathbf{w}^\top\boldsymbol{\Sigma}\mathbf{w}}})^2)] = \mathbf{0}, \tag{31}$$

and similarly,

$$\mathbf{v}_{-1}^* = -\frac{1}{2}\sum_{y\in\{1,-1\}}\boldsymbol{\mu}_{g_{y,-1}}. \tag{32}$$

Substitute the data assumption in Equation 30 and 32, we have

$$\mathbf{v}_a^* = [-a, 0, ..., 0]^\top. \tag{33}$$

We rewrite Equation 33 into a matrix product form:

$$\mathbf{v}_a^* = -\mathbf{P}\mathbb{E}[\mathbf{h}] = -\mathbb{E}[\mathbf{Ph}], \tag{34}$$

where $\mathbf{P} = \begin{pmatrix} 1 & 0 & \cdots & 0 \\ 0 & 0 & \cdots & 0 \\ \vdots & \vdots & \ddots & \vdots \\ 0 & 0 & \cdots & 0 \end{pmatrix}$.

We next verify local maximality. We compute the second derivative of $\mathcal{L}_{Acc}$ w.r.t. $\boldsymbol{v}_a$. Because the objective depend on $\boldsymbol{v}_a$ via the scalar projection, we define

$$s_a = \mathbf{w}^\top\boldsymbol{v}_a. \tag{35}$$

Rewrite the objective function of $s_a$

$$\mathcal{L}(s_a) = \frac{1}{2}\mathrm{erfc}(\frac{s_a + \mathbf{w}^\top\boldsymbol{\mu}_{g_{1,a}}}{\sqrt{2\mathbf{w}^\top\boldsymbol{\Sigma}\mathbf{w}}}) + \frac{1}{2}\mathrm{erf}(\frac{s_a + \mathbf{w}^\top\boldsymbol{\mu}_{g_{-1,a}}}{\sqrt{2\mathbf{w}^\top\boldsymbol{\Sigma}\mathbf{w}}}) + \frac{1}{2}. \tag{36}$$

To simplify notation, let

$$z_1(s_a) = \frac{s_a + \mathbf{w}^\top\boldsymbol{\mu}_{g_{1,a}}}{\sqrt{2\mathbf{w}^\top\boldsymbol{\Sigma}\mathbf{w}}}, \quad z_{-1}(s_a) = \frac{s_a + \mathbf{w}^\top\boldsymbol{\mu}_{g_{-1,a}}}{\sqrt{2\mathbf{w}^\top\boldsymbol{\Sigma}\mathbf{w}}}. \tag{37}$$

The second derivative w.r.t. $s$ as

$$\mathcal{L}''(s_a) = -\frac{2}{\sqrt{\pi}(2\mathbf{w}^\top\boldsymbol{\Sigma}\mathbf{w})}[z_1(s_a)\exp(-z_1(s_a)^2) - z_{-1}(s_a)\exp(-z_{-1}(s_a)^2)]. \tag{38}$$

At the stationary point, we have

$$z_1(s_a^*) = -z_{-1}(s_a^*). \tag{39}$$

Substitute back in equation 38, we obtain

$$\mathcal{L}''(s_a^*) = \frac{-2z_1(s_a^*)\exp(-z_1(s_a^*)^2)}{\sqrt{\pi}(\mathbf{w}^\top\boldsymbol{\Sigma}\mathbf{w})}. \tag{40}$$

The sign of equation 40 is determined by $z_1(s_a^*)$. We plug $\mathbf{v}_a^*$ in equation 37, we have

$$z_1(s_a^*) = \frac{\frac{1}{2}\mathbf{w}^\top(\boldsymbol{\mu}_{g_{1,a}} - \boldsymbol{\mu}_{g_{-1,a}})}{\sqrt{2\mathbf{w}^\top\boldsymbol{\Sigma}\mathbf{w}}}, \tag{41}$$

where $\mathbf{w}$ is the classifier for the positive sample, therefore $z_1(s_a^*) > 0$. Thus $\mathcal{L}''(s_a^*) < 0$. This indicates $\mathbf{v}_a^*$ is a local maximum.

Next, we compare the value of the objective $\mathcal{L}(s^*)$ with its value at the boundaries of the domain.

(i) As $s \to +\infty$, then $z_1(s_a) \to +\infty$ and $z_{-1}(s_a) \to +\infty$. Using the limits $\text{erfc}(-t) \to 2$ and $\text{erf}(-t) \to -1$ as $t \to +\infty$. We have $\mathcal{L}(s_a) = 1$.

(ii) As $s \to -\infty$, then $z_1(s) \to -\infty$ and $z_{-1}(s) \to -\infty$. Using the limits $\text{erfc}(-t) \to 0$ and $\text{erf}(-t) \to 1$ as $t \to -\infty$. We have $\mathcal{L}(s_a) = 1$.

(iii) At $s_a = s_a^*$, $\mathcal{L}(s_a^*) = 1 + \text{erf}(z_1(s_a^*))$, where $z_1(s_a^*) > 0$, it follows that

$$\mathcal{L}(s_a^*) > 1. \tag{42}$$

Thus the value of the objective function at $s_a^*$ is strictly greater than its value as $s_a \to \pm\infty$. The unique stationary point $s_a^*$ is the global maximizer, which concludes the statement $\square$.

## C  DERIVATION OF EQUATIONS 10 AND 11

**Modeling ROBOSHOT.**  ROBOSHOT is a method that linearly projects the image embedding onto the hyperplane associated with spurious features. Denote the spurious hyperplane as follows:

$$\mathbf{w}_a^\top\mathbf{x} = 0. \tag{43}$$

The projected point can be written as:

$$\mathbf{x}_{proj} = \mathbf{x} - \frac{\mathbf{w}_a^\top\mathbf{x}}{||\mathbf{w}_a||^2}\mathbf{w}_a. \tag{44}$$

Based on the spurious modeling 3.2, $\mathbf{h}$ follows a Gaussian mixture model. According to the relationship defined in Equation 44, each component in the Gaussian mixture model $\mathbf{x}_{proj} \sim \mathcal{N}(\boldsymbol{\mu}_{proj}, \Sigma_{proj})$ WorldSEnder, where

$$\boldsymbol{\mu}_{proj} = \boldsymbol{\mu} - \frac{\mathbf{w}_a^\top\boldsymbol{\mu}}{||\mathbf{w}_a||^2}\mathbf{w}_a, \tag{45}$$
$$\Sigma_{proj} = \boldsymbol{B}\Sigma\boldsymbol{B}^\top,$$

where $\mathbf{B} = \boldsymbol{I} - \frac{\mathbf{w}_a\mathbf{w}_a^\top}{||\mathbf{w}_a||^2}$, $\boldsymbol{\mu} = \mathbb{E}[\mathbf{x}]$. With Lemma 1, the analytical expression for ROBOSHOT is:

$$A_{ROBOSHOT}(\mathbf{h}, \mathbf{w}, \mathbf{w}_a; y) = \begin{cases} \frac{1}{2}\text{erfc}(-\frac{\mathbf{w}^\top(\boldsymbol{\mu} - \frac{\mathbf{w}_a^\top\boldsymbol{\mu}}{||\mathbf{w}_a||^2}\mathbf{w}_a)}{\sqrt{2\mathbf{w}^\top\mathbf{B}\Sigma\mathbf{B}^\top\mathbf{w}}}), & \text{if } y = 1 \\ \frac{1}{2}\text{erf}(-\frac{\mathbf{w}^\top(\boldsymbol{\mu} - \frac{\mathbf{w}_a^\top\boldsymbol{\mu}}{||\mathbf{w}_a||^2}\mathbf{w}_a)}{\sqrt{2\mathbf{w}^\top\mathbf{B}\Sigma\mathbf{B}^\top\mathbf{w}}}) + \frac{1}{2}, & \text{if } y = -1 \end{cases} \tag{46}$$

where $\mathbf{B} = \boldsymbol{I} - \frac{\mathbf{w}_a\mathbf{w}_a^\top}{||\mathbf{w}_a||^2}$.

**Modeling TIE.**  TIE is a method that translates the image embedding along the negative direction of the spurious vectors. With Lemma 1 and equation 8, the analytical expression for TIE is

$$A_{TIE}(\mathbf{h}, \mathbf{w}, \mathbf{w}_a; y) = \begin{cases} \frac{1}{2}\text{erfc}(-\dfrac{\mathbf{w}^\top(\boldsymbol{\mu} - \mathbf{w}_a^\top \boldsymbol{\mu}\mathbf{w}_a)}{\sqrt{2\mathbf{w}^\top \Sigma \mathbf{w}}}), \text{ if } y = 1 \\ \frac{1}{2}\text{erf}(-\dfrac{\mathbf{w}^\top(\boldsymbol{\mu} - \mathbf{w}_a^\top \boldsymbol{\mu}\mathbf{w}_a)}{\sqrt{2\mathbf{w}^\top \Sigma \mathbf{w}}}) + \frac{1}{2}, \text{ if } y = -1 \end{cases} \tag{47}$$

Next, plug the spurious feature classifier $\mathbf{w}_a = [1, \alpha, \mathbf{0}]$ and the label classifier $\mathbf{w} = [1, \beta, \mathbf{0}]$, and spurious data model in equation 46 and equation 47, we have

$$A_{ROBOSHOT}(\alpha, \beta; y) = \begin{cases} \frac{1}{2}\text{erfc}(-\dfrac{\alpha^2 - (1+\beta)\alpha + \beta}{\sqrt{2}(1+\alpha^2)(1+\alpha\beta)}), \text{ if } y = 1 \\ \frac{1}{2}\text{erf}(-\dfrac{\alpha^2 - (\beta-1)\alpha - \beta}{\sqrt{2}(1+\alpha^2)(1+\alpha\beta)}) + \frac{1}{2}, \text{ if } y = -1, \end{cases} \tag{48}$$

$$A_{TIE}(\alpha, \beta; y) = \begin{cases} \frac{1}{2}\text{erfc}(-\dfrac{(1+\beta)\sqrt{1+\alpha^2} - \alpha\beta - 1}{\sqrt{2(1+\beta^2)(1+\alpha^2)}}), \text{ if } y = 1 \\ \frac{1}{2}\text{erf}(-\dfrac{(1-\beta)\sqrt{1+\alpha^2} + \alpha\beta - 1}{\sqrt{2(1+\beta^2)(1+\alpha^2)}}) + \frac{1}{2}, \text{ if } y = -1, \end{cases} \tag{49}$$

The worst group accuracy takes the min value in equation 48 and equation 49.

## C.1  EXPERIMENT VALIDATION

Building on the theoretical analysis in Section 3.3, we further experimentally investigate the impact of various spurious classifiers on the worst group accuracy of TIE and ROBOSHOT. We generate 6 synonymous spurious text prompts using GPT 4 (OpenAI, 2023) for land features and 6 for water features, shown in Table 6. We test individual spurious text prompts, yielding 36 combinations (6 from water features, 6 from land features). The results are presented in Figure 6. Furthermore, we examine all possible combinations of two text prompts within the same spurious feature to expand the search range of spurious prompts, resulting in 225 combinations. These results are shown in Figure 7.

Table 6: Spurious Prompt used in experiments comparing ROBOSHOT and TIE.

| Spurious Template | Land Attributes | Water Attributes |
| --- | --- | --- |
| "A photo with a/an {a} background" | {land, field, hill, desert, forest, mountain} | {water, ocean, river, lake, sea, pond} |

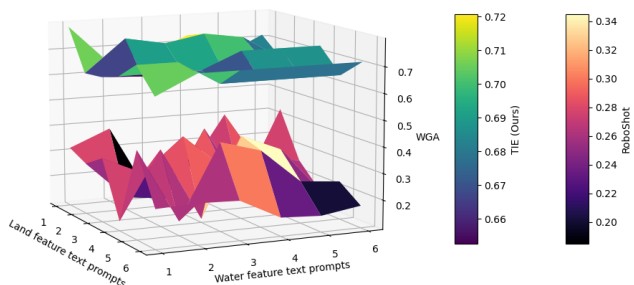

Figure 6: Experimental comparison between ROBOSHOT and TIE across different spurious text prompts, using a single spurious text prompt for each test.

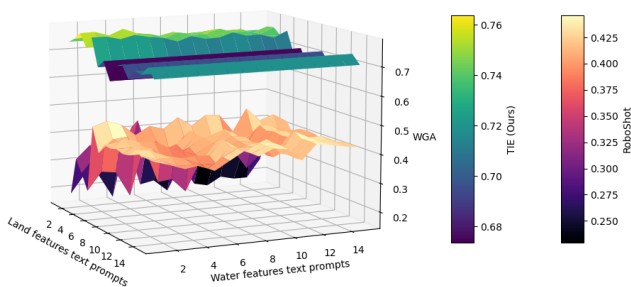

Figure 7: Experimental comparison between ROBOSHOT and TIE on different spurious text prompts, using multiple spurious text prompts for each test.

From Figure 6 and 7, we observe a significant performance gap between TIE and ROBOSHOT. This suggests that TIE is more robust, and less dependent on the accuracy of spurious text prompts compared to ROBOSHOT.

## D  ALGORITHM FOR TIE⋆

---
**Algorithm 1** TIE⋆

---
**Input:** Input $\mathbf{x}$, Image encoder $\phi_I(\cdot)$, Text encoder $\phi_T(\cdot)$, Spurious text prompts $\mathcal{T}_a$, Target text prompts $\mathcal{T}$.
**Output:** Predicted label $\hat{y}$.

1: **for** $\mathbf{t}_a \in \mathcal{T}_a$ **do**
2:      $\mathbf{v}_a = \phi_T(\mathbf{t}_a)$                                  ▷ Computing the spurious vector
3:      $\mathbf{v}_a = \frac{\mathbf{v}_a}{||\mathbf{v}_a||}$                                            ▷ Normalize
4: **end for**
5: $\hat{a} = \arg\max_{a \in \mathcal{A}} < \phi_I(\mathbf{x}), \phi_T(\mathbf{t}_a) >$          ▷ Psuedo labeling on spurious feature
6: $\mathbf{h}_{\hat{a}} = \phi_I(\mathbf{x}; \hat{a})$                                        ▷ Image embedding
7: $\lambda_{\hat{a}} = \mathbb{E}[(\mathbf{h}_{\hat{a}}^{\top} \mathbf{v}_{\hat{a}})]$                            ▷ Estimate the scale coefficient
8: $\mathbf{h}_{\hat{a}} \leftarrow \mathbf{h}_{\hat{a}} - \lambda_{\hat{a}} \mathbf{v}_{\hat{a}}$                             ▷ Translate image embedding
9: $\hat{y} = \arg\max_{y \in \mathcal{Y}} < \mathbf{h}_{\hat{a}}, \phi_T(\mathbf{t}_y) >$              ▷ Zero shot classfication
10: **return** $\hat{y}$

---

### D.1 ESTIMATION OF THE SCALE COEFFICIENT $\lambda_a$

Based on Equations 30 and 32, the theoretical optimal scale coefficient can be computed as

$$\lambda_a = \frac{1}{|\mathcal{Y}|} \sum_{y \in |\mathcal{Y}|} \mathbb{E}[(\mathbf{h}_a^\top \mathbf{v}_{a,y})], \tag{50}$$

where $|\mathcal{Y}|$ denotes the number of classes. However, directly computing equation 50 is intractable since the component $\mathbf{v}_{a,y}$ requires label information, which violates the zero-shot setting. To address this issue, we analyze the distributions of the two groups: $\mathbf{h}_a^\top \mathbf{v}_{a,y}|a = 1, y = -1$ and $\mathbf{h}_a^\top \mathbf{v}_{a,y}|a = 1, y = 1$. We perform a two-sample $t$-test under the null hypothesis $\mathcal{H}_0$ that the means of these two distributions are equal. Experimental results on the Waterbirds dataset yield $p = 0.78$ for ViT/B and $p = 0.40$ for ViT/L, both of which are greater than the significance level of $p = 0.05$. Hence, we do not have sufficient evidence to reject $\mathcal{H}_0$. In other words, the mean values of the two distributions are equal.

Figure 8 illustrates the density distributions of the scale coefficients. This statistical test indicates that the mean scale coefficient is consistent across groups. Therefore, we can estimate the scale coefficient as

$$\lambda_a = \mathbb{E}[(\mathbf{h}_a^\top \mathbf{v}_a)]. \tag{51}$$

We further validate our estimated scale coefficient on the Waterbirds dataset using both the ViT/B and ViT/L backbones. We iteratively adjust the scale coefficient and plot the worst group accuracy for each scale value. Figure 9 shows that the estimated scale is nearly identical to the theoretical optimal scale. Moreover, the worst group accuracy obtained using the estimated scale demonstrates robust performance.

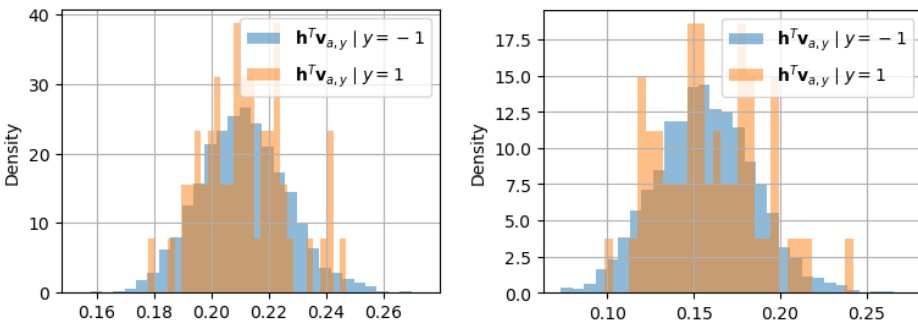

Figure 8: The distribution of $\mathbf{h}_a^\top \mathbf{v}_{a,y}$ across different target groups. *Left:* Results from testing with ViT/B. *Right:* Results from testing with ViT/L.

## E  DATASET

We evaluate our method and all comparison methods on the following datasets:

- **Waterbirds** (Koh et al., 2021; Sagawa et al., 2019): The primary task of the Waterbirds dataset is to classify bird types, specifically, $y = \{$Landbird, Waterbird$\}$. The spurious confounder in this dataset is the background, $a = \{$Land background, Water background $\}$. It includes four groups: {Landbird with a Land background, Landbird with a Water background, Waterbird with a Land background, Waterbird with a Water background}.

- **CelebA** (Liu et al., 2015): The CelebA dataset comprises over 200K celebrity faces. Following the protocol by (Sagawa et al., 2019), the task is to identify hair color with target labels $y = \{$dark hair, blonde hair$\}$. The spurious correlation label is gender, $a = \{$female, male$\}$. This dataset is segmented into four groups: {a female with dark hair, a female with blonde hair, a male with dark hair, a male with blonde hair}.

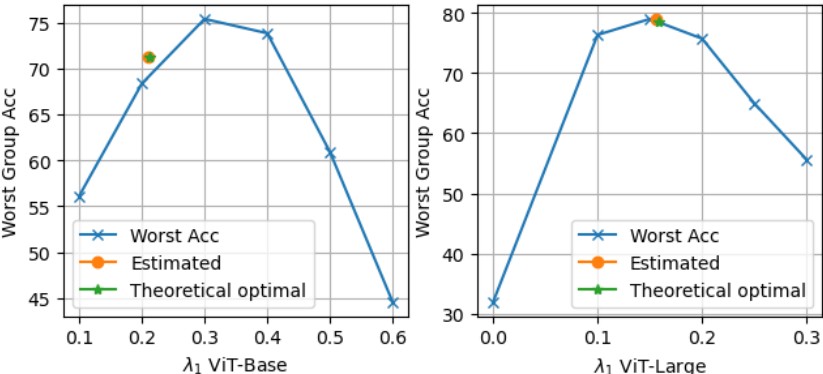

Figure 9: Relationship between worst-group accuracy and the scale coefficient. The estimated scale coefficient closely aligns with the theoretical optimal value and approaches the maximum achievable worst-group accuracy.

- **ISIC** (Codella et al., 2019): The ISIC dataset is utilized for skin cancer diagnosis. Following the task from (Wu et al., 2023), the task is to predict the type of skin cancer, denoted as $y = \{$Benign, Malignant$\}$. The spurious correlation feature in this dataset is $a = \{$with color patch, without color patch$\}$. It encompasses three groups: $\{$Benign cancer with a color patch, Benign cancer without a color patch, Malignant cancer without a color patch$\}$.

- **COVID-19** (Cohen et al., 2020): The COVID-19 dataset is used to diagnose from X-ray images, with the classification task defined as $y = \{$no Pneumonia, pneumonia$\}$. The spurious confounder in this dataset is gender, $a = \{$male, female$\}$. It consists of four groups: $\{$a male with pneumonia, a male without pneumonia, a female with pneumonia, a female without pneumonia$\}$.

- **FMOW** (Christie et al., 2018): The Functional Map of the World (FMOW) is a large-scale satellite image dataset comprising 62 classes. We follow the protocol outlined in (Wu et al., 2023; Izmailov et al., 2022) to define groups based on geographical regions: Africa, the Americas, Oceania, Asia, and Europe.

## F  BASELINES

We compare `TIE` against several state-of-the-art methods for zero-shot classification.

- **Group Prompt**: Group Prompt is a method that includes spurious correlation labels in text prompts. For example, in the waterbirds dataset, the text prompts for Group Prompt specify the background along with the bird type, [`a photo of a landbird with land background, a photo of a landbird with a water background, a photo of a waterbird with a land background, a photo of a waterbird with a water background`].

- **Ideal words** (Trager et al., 2023): The ideal prompt is to start by adding prompts related to target labels before integrating those associated with spurious correlation attributes. Subsequently, the ideal method averages across all the spurious correlation prompts.

- **Orth-Cali** (Chuang et al., 2023): The Orth-Cali method is designed to debias text prompts by making the text embeddings invariant to spurious features. This approach introduces a projection matrix that projects the text into the null space defined by the span of spurious text prompts. It then employs regularization to ensure that these projected prompts are closely mapped within the text embedding space.

- **Perception CLIP** (An et al., 2024): Perception CLIP is a method inspired by empirical findings that suggest that including contextual attributes in text prompts enhances zero-shot classification performance and mitigates the effects of spurious correlations. To improve the group robustness, Perception CLIP incorporates information about spurious features.

- **ROBOSHOT** (Adila et al., 2024): Roboshot is a method that utilizes LLMs to identify spurious insights. It then removes these spurious features from the image embeddings using the Gram-Schmidt process, which projects the image embeddings onto a space orthogonal to that of the spurious insights. Subsequently, Roboshot enhances the image embeddings by projecting them along vectors representing helpful insights.

## G  IMPLEMENTATION

We conducted all experiments on an Nvidia RTX 3090 GPU with 24 GB of memory, using frozen CLIP models across various datasets. Specifically, for the Waterbirds and CelebA datasets, the vision encoder backbones included ViT-B-32 (Dosovitskiy et al., 2020), ViT-L-14 (Dosovitskiy et al., 2020), and ResNet 50 (He et al., 2016). Model construction and pre-trained weights are sourced from Open CLIP (Ilharco et al., 2021).

For specialized datasets, including ISIC and COVID-19, we employed the Biomed CLIP backbone (Zhang et al., 2023a), acknowledging that the training set from general CLIP significantly diverges from the biomedical context, leading to substantial shifts in test performance. With ViT-L-32, we observed 0 % worst-group accuracy. Hence, we excluded results using the general backbone for these specialized datasets.

As no training was conducted for all methods, the results are deterministic.

To facilitate the reproduction of our results, we have detailed both the label prompts and spurious prompts in Table 7. Note that the nature of CLIP is sensitive to prompts; our spurious prompts are created through simple adaptations of the label prompts. We incorporate our label prompts and spurious prompts in all comparison methods except for vanilla zero-shot to ensure a fair comparison.

Table 7: Prompts details

| **Dataset** | Label prompts | Spurious prompts |
|---|---|---|
| Waterbirds | [a photo of a landbird, a photo of a waterbird] | [a photo with a water background, a photo with a land background] |
| CelebA | [a photo of a celebrity with dark hair, a photo of a celebrity with blonde hair] | [female, male] |
| ISIC | [This is a benign lesion, This is a malignant lesion] | [There exists no color patch, There exists a color patch] |
| COVID-19 | [An X-ray image of a chest without Pneumonia, An X-ray image of a chest with Pneumonia] | [An X-ray image from a female, An X-ray image from a male] |

## H  ABLATION STUDY

### H.1  DIFFERENT SPURIOUS TEXT PROMPT TEMPLATES

Beyond the textual description of spurious features, the format of spurious text prompt templates also impacts the performance. To further validate the effectiveness of all methods, we conducted experiments using various text templates, including '{spurious feature label}' and 'A photo with a spurious feature, {spurious feature label}, in the waterbirds dataset. The results are presented in Table 9.

Table 8: FMOW Prompt details

| Class-Template | Spurious Template | Class $y$ | Group $g$ |
|---|---|---|---|
| A satellite image of a/an $\{y\}$. | Over $\{g\}$ | {airport, airport hangar, airport terminal, amusement park, aquaculture, archaeological site, barn, border checkpoint, burial site, car dealership, construction site, crop field, dam, debris or rubble, educational institution, electric substation, factory or powerplant, fire station, flooded road, fountain, gas station, golf course, ground transportation station, helipad, hospital, impoverished settlement, interchange, lake or pond, lighthouse, military facility, multi-unit residential, nuclearpowerplant, office building, oil or gas facility, park, parking lot or garage, place of worship, police station, port, prison, race track, railway bridge, recreational facility, road bridge, runway, shipyard, shopping mall, single-unit residential, smokestack, solar farm, space facility, stadium, storage tank, surface mine, swimming pool, toll booth, tower, tunnel opening, waste disposal, water treatment facility, wind farm, zoo} | {Europe, Asia, Americas, Africa, Oceania} |

Table 9: Zero-shot classification results on the Waterbirds dataset with different spurious prompt templates. T1: {Spurious feature label}, T2: A photo with a spurious feature, {Spurious feature label}. (CLIP ViT-B/32)

| | T1 Spurious Template | | | T2 Spurious Template | | |
|---|---|---|---|---|---|---|
| | WG ↑ | Avg ↑ | Gap ↓ | WG ↑ | Avg ↑ | Gap ↓ |
| ZS | 41.37 | 68.48 | 27.11 | 41.37 | 68.48 | 27.11 |
| Group Prompt | 43.46 | 66.79 | 23.33 | 43.46 | 66.79 | 23.33 |
| Ideal words | 61.99 | 78.87 | 16.88 | 60.44 | 79.82 | 19.38 |
| Orth-Cali | 64.08 | 73.74 | 9.66 | 67.14 | 76.58 | **9.44** |
| Perception CLIP | 23.37 | 61.54 | 38.17 | 46.20 | 73.37 | 27.17 |
| ROBOSHOT | 44.35 | 69.03 | 24.68 | 45.99 | 69.67 | 23.68 |
| TIE | **71.04** | **80.11** | **9.07** | **69.63** | **82.02** | 12.39 |
| TIE* | 56.14 | 75.00 | 18.86 | 67.60 | 79.84 | 12.24 |

## H.2 MORE BACKBONE RESULTS.

Our paper focuses on CLIP as it serves as a foundational model widely applied across various domains, like in stable diffusion (Rombach et al., 2022). Beyond the CLIP family models, we have expanded our experiments to incorporate various backbone models. We utilize ALIGN (Jia et al., 2021) backbones on the Waterbirds dataset, with results shown in Table 10.

From Table 9 and 10, we observe that TIE demonstrates robust performance across various spurious prompt templates and different backbones, indicating significant potential for real-world applications.

Table 10: Zero Shot classification results on the Waterbirds dataset with the ALIGN backbone

|  | WG ↑ | Avg ↑ | Gap ↓ |
|---|---|---|---|
| ZS | 47.50 | 69.83 | 22.33 |
| Group Prompt | 5.81 | **72.55** | 66.74 |
| Ideal words | 51.71 | 67.17 | 15.46 |
| Orth-Cali | 28.35 | 58.73 | 30.38 |
| Perception CLIP | 31.60 | 54.39 | 22.79 |
| ROBOSHOT | 41.02 | 50.95 | **9.93** |
| TIE | **56.07** | 69.54 | 13.47 |
| TIE* | 52.49 | 64.27 | 11.78 |

## I  DISCUSSION ON TEXT PROMPTS

The effectiveness of VLMs depends on the quality of text prompts. The guidelines for selecting text prompts represent a critical area for deeper exploration. To address this, we show our insights through experiments designed to identify an effective and generalizable approach for creating optimal text prompts in practice.

We investigate this issue by decomposing a text prompt into a template and an object.

- T1: "A photo with [Object]"

- T2: "A photo with a spurious feature, [Object]"

- T3: "[Object]"

For the object, Ge et al. (2023) shows that labels exhibit a hierarchical structure in "WordNet" Fellbaum (1998). For example, the hierarchical progression of the word 'strawberry' includes 'berry', 'edible fruit', 'food', each level becoming more general Ge et al. (2023). In our experiments, we test three labeling strategies: using the level directly above to represent a more generalized category, the spurious feature itself, and an average of the top five most specific terms at the bottom of the hierarchy for greater specificity. We provide details of the object candidates in Table 11. The aim is to determine the most effective level of generality or specificity for descriptions. We conducted experiments on the Waterbirds dataset using TIE* (ViT-L14). The results are shown in Table 12.

Table 11: Object candidates

|  | Water background prompts | Land background prompts |
|---|---|---|
| O1 (hypernyms) | Fluid | Ground |
| O2 (self) | Water | Land |
| O3 (hyponyms) | Sea, Lake, River, Stream, Creek | Arable Land, Farmland, Forest Land, Grassland, Desert |

*Insights:* We note that using a proper object description is important. We suggest using a specific description of the spurious feature or their hyponyms, as this can improve the worst group accuracy (WG) in the zero-shot classification. In contrast, using overly general descriptions such as hypernyms significantly degrades performance. This observation aligns with recommendations for specificity and clarity in text prompt engineering for language models Ekin (2023).

In terms of templates, we found that giving a portion of contextual information, such as the prefix "a photo with" or "a photo with a spurious feature," helps the WG. Templates lacking a prefix demonstrate poor performance, a finding that aligns with the observations presented in Radford et al. (2021). For practical purposes in ViT-based CLIP models, we encourage users to adopt templates that include a prefix, with the object description utilizing the spurious feature itself, balancing ease of use and performance.

Table 12: Performance evaluation of CLIP-ViTL14 for `TIE*`, We have highlighted in bold the results that surpass the WG in Table 1.

| Text prompts | WG ↑ | Avg ↑ | Gap ↓ |
|---|---|---|---|
| T1+O1 | 53.97 | 76.49 | 22.52 |
| T1+O2 | 61.60 | 78.98 | 17.38 |
| T1+O3 | **65.26** | 80.20 | 14.94 |
| T2+O1 | 46.48 | 72.69 | 26.21 |
| T2+O2 | **63.77** | 80.35 | 16.58 |
| T3+O3 | **63.19** | 79.06 | 15.87 |
| T3+O1 | 45.90 | 73.19 | 27.29 |
| T3+O2 | 60.62 | 78.84 | 18.22 |
| T3+O3 | 59.56 | 77.91 | 18.35 |

## J    Future direction discussion

We introduce `TIE` to mitigate the effect of spurious correlations, which are vital in prediction tasks. While our approach demonstrates strong performance, it faces challenges redirecting attention to the object in the presence of pronounced artifacts (e.g., watermarks) without appropriate text prompts. Figure 10 illustrates a rare case where the dominant feature is a watermark. To evaluate our method's capability in redirecting attention, we provide the following text prompts:

- Text prompt 1 (TP1): `A photo with a water background,`
- Text prompt 2 (TP2): `A photo with a watermark`

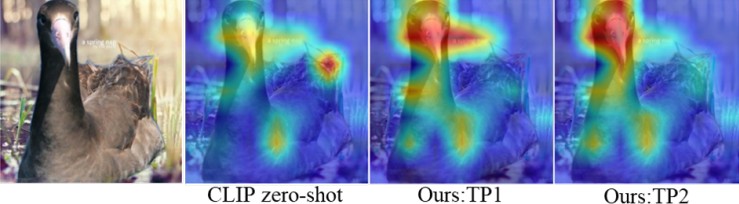

CLIP zero-shot      Ours:TP1      Ours:TP2

Figure 10: Attention-based explanations in an image with a strong artificial landmark in the Waterbirds dataset. TP1: `A photo with a water background,` TP2: `A photo with a watermark.`

From Figure 10, we observe that when using TP1, a text prompt representing a common spurious feature in the dataset, the attention fails to redirect back to the correct core feature (the bird in the image). Interestingly, when providing a corresponding text prompt (TP2), the attention successfully shifts from the watermark to the bird. This highlights the potential of our proposed method to address misclassifications caused by factors beyond spurious correlations, offering a promising direction for further research.

## K    Failure Case analysis for TIE*

`TIE*` is a method free from using any annotations and requires the spurious text prompt for inference of the spurious label in the dataset. We analyzed TIE* failure cases, which can be broadly categorized into two scenarios: (1) inaccuracies in the pseudo-spurious labels and (2) images containing artifacts (e.g., watermarks).

For (1): The majority of failures in TIE* occur when zero-shot classification incorrectly assigns a spurious label. This misassignment causes samples to be translated in the opposite direction, leading to incorrect classifications. In Section 4.4, we examine the worst-group accuracy in zero-shot classification and the accuracy of pseudo-spurious labels. Our analysis reveals that the pseudo-spurious

labels assigned by TIE* have a direct impact on the worst-group accuracy in zero-shot classification: higher accuracy in assigning these labels corresponds to improved worst-group accuracy.

To potentially improve `TIE*`'s performance, we propose three practical strategies: utilizing group-robustified spurious text prompts (Section 4.3), employing a small subset of spurious-labeled data (Section 4.4), and following the guidelines for effective text prompts (Section I) to achieve better performance.

For (2): we discussed this scenario in Section J. This is a case where the artifact (e.g., a watermark) becomes the dominant feature. While using `TIE` or `TIE*` reduce dependency on spurious features (such as background information), it cannot eliminate the effect of the artifact. This limitation can lead to the failure of our algorithm. Interestingly, we found that `TIE` has the potential to remove unwanted features when provided with appropriate text prompts. However, the identification of these incorrect features remains an open area for further investigation.

## L    BROADER IMPACTS

Our work aims to mitigate spurious correlations in VLM models, a crucial endeavor for the machine learning community. Beyond enhancing group robustness, the positive impacts of our work extend to domains such as fairness, trustworthiness, and generalization. This is particularly significant when deploying machine learning algorithms in high-stakes domains.

