# OpenReview forum: "Mitigating Spurious Correlations in Zero-Shot Multimodal Models"
_ICLR.cc/2025/Conference — ICLR 2025 Poster_

### Official Review · Reviewer_gGse · 2024-10-30

**Soundness:** 3
**Presentation:** 3
**Contribution:** 3
**Rating:** 6
**Confidence:** 4

**Summary:**

To address the spurious correlations existing in VLMs, the authors propose a new solution called TIE  that tackles spurious correlations within the zero-shot setting without requiring target label annotations, which would contradict the zero-shot principle. TIE utilizes a translation operation in the latent space that preserves the distribution of image embeddings, guided by text prompts to address spurious correlations. This approach is theoretically grounded, identifying optimal translation directions along the spurious vector.

**Strengths:**

1. The experiments in this paper are comprehensive and have been verified on datasets from many different fields (four different image domains), proving that the method is universal and has good generalization properties.

2. The structure of the paper is clear, and the methods proposed are easy to follow.

**Weaknesses:**

1. The performance improvements of the proposed method are not particularly obvious.

**Questions:**

1. When RESNET-50 is used as the backbone, the proposed method performs poorly. Is there a way to solve the misalignment between text and image?

2. How do we ensure the accuracy of the textual hints provided when using artificially identified fake features? Is there an evaluation mechanism to verify the effectiveness of these hints?

---

> ### Author Response · Authors · 2024-11-21
> **To our reviewer gGse**
>
> We sincerely appreciate your positive feedback and thorough review. Your question holds great importance to us, and we would like to take this opportunity to address it.
>
> >(1) When RESNET-50 is used as the backbone, the proposed method performs poorly. Is there a way to solve the misalignment between text and image?
>
> **Q(1)** For a fair comparison, we employed the same text prompts across models such as ViT and ResNet. However, we found that ResNet-50 shows potential misalignment between text and image embeddings.
>
> We hypothesize that this misalignment stems from differences in expressive capabilities between the text and image modalities. Specifically, the ResNet-50 encoder shows reduced expressiveness in its representations.
>
> To verify this hypothesis and address the misalignment, we conducted experiments using text prompts with reduced contextual information to emphasize the main object. This approach is inspired in the mechanics of text embeddings, which are computed via attention mechanisms. By reducing contextual information, attention is more focused on the main object, thereby reducing redundancy in the text embeddings. Such reduction is critical, as excessive information can overwhelm less expressive models like ResNet-50.  We run experiments on the following spurious text prompts for both TIE and TIE*.
>
>
> Table 1. Zeroshot classification results on Waterbirds with ResNet-50 for TIE.
> |Spurious text prompts|WGA|Acc|
> |-|-|-|
> |"a photo with a water background", "a photo with a land background"| 52.96 | 83.62  |
> |"a water background", "a land background"|59.50 |  83.34 |
> |"water", "land"| 65.58 | 82.76 |
>
> Table 2. Zeroshot classification results on Waterbirds with ResNet-50 for TIE*.
> |Spurious text prompts|WGA|Acc|
> |-|-|-|
> |"a photo with a water background", "a photo with a land background"| 34.11 | 81.19  |
> |"a water background", "a land background"| 43.14  | 80.91   |
> |"water", "land"| 63.55  | 77.91  |
>
> From Tables 1 and 2, we observe that using shorter spurious text prompts improves the worst-group accuracy, significantly enhancing performance. This finding contradicts the recommended practices for text prompts in ViT-based models, as discussed in A1 of the general response. This further validates the existence of misalignment between text and image embeddings in ResNet-based models, which arises from differences in their underlying backbone architectures.
>
>
> >(2) How do we ensure the accuracy of the textual hints provided when using artificially identified fake features? Is there an evaluation mechanism to verify the effectiveness of these hints?
>
> **Q(2)** The reliability of spurious text prompts is a significant challenge. The strength of our proposed method lies in its robustness, as it relies less on highly accurate spurious vectors compared to existing work.  For further discussion on this method, we kindly refer our reviewer gGse to Section A1 of the general response.
>
> Evaluating the effectiveness of text prompts is a critical aspect of multimodal model research. However, in our zero-shot problem setting, we do not have access to any target label information, making the evaluation of text prompts challenging. Nonetheless, we present our recommendations for handling spurious text prompts in our general response A1, which have been empirically validated. Given the less sensitive nature of our method, combined with best practices for constructing text prompts, it can effectively mitigate spurious correlations in the zero-shot classification of VLMs.

---

### Official Review · Reviewer_wW81 · 2024-10-30

**Soundness:** 4
**Presentation:** 4
**Contribution:** 4
**Rating:** 8
**Confidence:** 4

**Summary:**

The paper proposes a way to robustify VLMs like CLIP against spurious correlation. The proposed method (TIE) does not require any extra data and fine-tuning; and operate across modalities (as opposed to previous method that only operates in single modalities). In order to preserve the image embedding distribution, the proposed method is based on a translation operation in the embedding space, and find the optimal translation vector based on theoretical analysis. In practice, this translating vector is found using spurious text prompts.

Theoretically, the authors show how this method compare to another zero-shot rebustification method, Roboshot; and show the major drawback of Roboshot that is tackled by TIE. Empirically, the method outperforms almost all existing baselines.

**Strengths:**

- Very nice theoretical analysis that backs the proposed methods. First the authors derives what is the optimal translation vector based on the definition of zero-shot classification and group robustness problems.
- Very nice comparison with RoboShot, which strongly shows why the proposed method is necessary to improve upon RoboShot.
- Strong empirical results with strong baselines.
- Nice visualization results in 4.5, strongly backs the claim why CLIP robustification is necessary in the first place.

**Weaknesses:**

1. Some notations need to be clarified (check Questions).
2. I think the TIE* procedure needs to be clarified more, do we need to know possible values of $t_a$ beforehand? this need to be made clearer in the early in the writing when TIE* is introduced.

**Questions:**

1. What is $h_a$ in Theorem 1? is it image embeddings w/ spurious features $a$ regardless of the label?
2. What is the difference between $t_{spu}$ and $t_a$? (Equation 9)
3. For TIE*, does it mean we need access to all possible spurious features descriptions? Not necessarily a drawback as mostly in datasets we know this beforehand

---

> ### Author Response · Authors · 2024-11-21
> **To our reviewer wW81**
>
> We sincerely appreciate your inspiring review and the valuable questions you've raised. We would like to take this opportunity to respond thoughtfully to each of them.
>
> *Weakness:*
> >(1)Some notation need to be clarified
>
> **W(1)** We will address this point in our responses within the question section.
>
> >(2) I think the TIE* procedure needs to be clarified more, do we need to know possible values of $t_a$ beforehand? this need to be made clearer in the early in the writing when TIE* is introduced.
>
> **W(2)** Thank you for your suggestion, we will clarify this in our revision. $t_a$ is a predetermined spurious text prompt, with all possible values should be known beforehand. In our experiments, TIE* employs these spurious text prompts for zero-shot classification to assign pseudo-spurious labels. We will ensure this is explicitly stated in the revised manuscript.
>
> *Question*
> >(1) What is $h_a$ in Theorem 1? is it image embeddings w/ spurious feature $a$ regardless of the label?
>
> **Q(1)** Yes, $h_a$ is the image embedding with spurious feature $a$ regardless the label.
>
> >(2) What is the difference between $t_{spu}$ and $t_a$?
>
> **Q(2)** Thank you for pointing out the interchangeable use of the subscripts $a$ and ${spu}$. Both notations are intended to denote the same concept: ${spu}$ refers to the spurious text prompt, and $a$ to the corresponding spurious label. We will ensure consistent notation in the revised version of our manuscript. Thank you!
>
> >(3): For TIE*, does it mean we need access to all possible spurious features descriptions? Not necessarily a drawback as mostly in datasets we know this beforehand.
>
> **Q(3)** Yes, for TIE*, accessing all possible descriptions of spurious features is required. In experiments, the same spurious text prompts are used both to compute pseudo-spurious labels and to guide the translation of image embeddings.

---

> > ### Comment · Reviewer_wW81 · 2024-11-23
> > **Response to authors**
> >
> > Thank you for your response! I have no further questions

---

> > > ### Author Response · Authors · 2024-11-23
> > > **Thank you note**
> > >
> > > Dear Reviewer wW81,
> > >
> > > We sincerely appreciate your invaluable support for our manuscript. Your inspiring review holds significant value to us, and we are deeply grateful for your insightful feedback. Should you have any additional comments, questions, or suggestions, we would be truly honored to address them. Once again, thank you for your careful and thorough review of our work!
> > >
> > > With warm regards,
> > >
> > > Authors

---

### Official Review · Reviewer_RFvW · 2024-11-01

**Soundness:** 3
**Presentation:** 3
**Contribution:** 3
**Rating:** 6
**Confidence:** 3

**Summary:**

- Paper proposes TIE, a method to mitigate spurious correlations in VLMs.
- TIE uses a translation operation in the latent space, guided by spurious text prompts, to shift image embeddings away from spurious features.
- The approach preserves the embedding distribution, unlike projection-based methods.
- TIE*, a variant, uses the VLM itself to infer spurious labels when explicit labels are unavailable.
- Experiments on several benchmarks show improvements over other methods.

**Strengths:**

- TIE/TIE*'s translation-based approach is a novel contribution to the field as far as I know.
- The method has strong theoretical analysis
- It effectively addresses spurious correlations without requiring training or labeled data.
- The experimental results are good.

**Weaknesses:**

- TIE relies on accurate spurious prompts. The authors propose TIE* but its performance is not as good as TIE. Further investigation into the limitations of TIE* and potential improvements would help.
- The experiments focus on CLIP-based models. Exploring TIE/TIE* on other VLMs would be helpful.
- nitpick: the capitalization of the term "TIE" implies its an acronym. If it is an acronym, it's unclear what it stands for.

**Questions:**

1. How sensitive is TIE to the choice of spurious prompts? Are there any guidelines for selecting effective prompts?
2. What are the failure cases of TIE* and how can automatic inference of spurious labels be improved?

---

> ### Author Response · Authors · 2024-11-21
> **To our reviewer RFvW**
>
> Thank you for your considerate and encouraging review. We appreciate the opportunity to address your insightful question.
>
> *Weakness:*
> >(1)TIE relies on accurate spurious prompts. The authors propose TIE* but its performance is not as good as TIE. Further investigation into the limitations of TIE* and potential improvements would help.
>
> **W(1)** We kindly redirect our reviewer to Q1 in our response to Reviewer Y3Qc.
>
> >(2) The experiments focus on CLIP-based models. Exploring TIE/TIE* on other VLMs would be helpful.
>
> **W(2)**: Due to page constraints, we have included experiments with additional VLMs in the appendix of our original submission. In Section H.2, we conducted experiments using the ALIGN backbone [1] to perform zero-shot classification tasks. We observed that our method consistently achieves good performance in terms of worst group accuracy. This demonstrates the robustness of our method across different types of VLMs.
>
> >(3) nitpick: the capitalization of the term "TIE" implies its an acronym. If it is an acronym, it's unclear what it stands for.
>
> **W(3)** Thank you for pointing this out. TIE stands for **T**ext prompt based **I**mage **E**mbedding translation, which simplifies references in the paper.
>
> *Question:*
>
> >(1) How sensitive is TIE to the choice of spurious prompts? Are there any guidelines for selecting effective prompts?
>
> **Q(1)** We kindly redirect our reviewer to A1 section in general response.
>
> >(2) What are the failure cases of TIE* and how can automatic inference of spurious labels be improved?
>
> **Q(2)** We kindly redirect our reviewer to A2 section in general response.
>
> [1] Jia, Chao, et al. "Scaling up visual and vision-language representation learning with noisy text supervision." International conference on machine learning. PMLR, 2021.

---

> > ### Comment · Reviewer_RFvW · 2024-11-23
> >
> > Thank you for your response. I have no further questions.

---

> > > ### Author Response · Authors · 2024-11-23
> > > **Thank you once again**
> > >
> > > Dear Reviewer RFvW,
> > >
> > > Thank you very much for your insightful and constructive reviews. We are truly pleased to have addressed your questions. If you have any additional questions or need further clarification, we would be more than happy to provide them. Once again, we extend our heartfelt gratitude for your thoughtful review of our manuscript.
> > >
> > > With warm regards,
> > >
> > > Authors

---

### Official Review · Reviewer_y3Qc · 2024-11-05

**Soundness:** 3
**Presentation:** 2
**Contribution:** 3
**Rating:** 6
**Confidence:** 3

**Summary:**

In this paper, the authors proposed a method to address spurious correlations in Vision Language Models while preserving their zero-shot capabilities. Drawing on theoretical analysis to explore optimal strategies for translating image embeddings, they introduce TIE—a simple algorithm that guides the translation of image embeddings based on the text prompt. The core idea is  to apply a translation transformation in the latent space using "spurious vectors" derived from text prompts to better align text and image embeddings. Experimental results on real-world datasets demonstrate that this approach not only improves the worst-group accuracy across all datasets but also achieves comparable overall accuracy, enhancing the robustness of VLMs across modalities.

**Strengths:**

- The important task of mitigating spurious correlations in VLMs
- The interesting idea of applying a translation transformation in the latent space for cross-modal alignment
- Rigorous theoretical analysis to examine optimal strategies for translating image embeddings
- Quantitative and qualitative results validating the effectiveness of the proposed method

**Weaknesses:**

- More in-depth discussion of the method is necessary (Why does it work? When does it fail? etc.)
- English can be improved

**Questions:**

- Have you conducted an evaluation of the quality of the predicted pseudo-spurious labels on a validation set in order to analyze and bridge the accuracy gap between TIE and TIE*?

- In large-scale datasets and tasks, how valid is the assumption of a single spurious feature for the entire dataset? How can spurious features be efficiently identified on a per-data-point basis?

---

> ### Author Response · Authors · 2024-11-21
> **To our reviewer y3Qc part 1**
>
> we sincerely appreciate your positive feedback and thorough review. We hope to take this opportunity to address your questions.
>
> *Weakness*
> >(1) More in-depth discussion of the method is necessary (Why does it work? When does it fail? etc.)
> >
> **W(1)** [Why does it work?] Our work is grounded in a theoretical analysis which establishes that, under conditions of spurious correlations, the optimal translation operator for image embeddings aligns in the opposite direction of the spurious vector. Conceptually, this can be interpreted as neutralizing the influence of spurious features. In the domain of VLMs, the spurious vector can be effectively identified through the use of spurious text prompts. Building on this foundation, we propose TIE and TIE* as solutions to address spurious correlations in VLMs. Our method effectively integrates both image and text modalities, satisfying the requirements of VLMs and is supported by a solid theoretical foundation. These factors collectively provide a strong rationale for why our methods consistently achieve the best or second-best WGA performance across various datasets.
>
> [When does it fail?] We kindly redirect Reviewer Y3Qc to the general response A2 for detailed information.
>
>
> >(2) English can be improved.
>
> **W(2)** Thank you for your suggestion. We will refine our wording in the revised version.
>
> *Question*
> >(1) Have you conducted an evaluation of the quality of the predicted pseudo-spurious labels on a validation set in order to analyze and bridge the accuracy gap between TIE and TIE*?
>
> **Q1**. In Section 4.4, we investigated the accuracy of pseudo-spurious labels and their relationship with the worst group accuracy (WGA) of the zero-shot classification, as shown in Figure 4 on the right-hand side. We demonstrated that the accuracy of predicting spurious labels monotonically benefits the WGA in zero-shot classification. To further validate this relationship, we conducted experiments on the CelebA dataset, where we computed the accuracy for the spurious label 'male' using various models: CLIP ViT-B32, CLIP ViT-L 14, and CLIP ResNet-50. The results are shown below.
>
> Table 1. Relationship between pseudo spurious label accuracy and performance Gap between TIE and TIE*.
> ||Suprious label Acc|WGA Gap between TIE and TIE*|
> |-|-|-|
> |CLIP(ViT-B32)| 98.67 | 0.02  |
> |CLIP(ViT-L14)| 94.60 |  2.62 |
> |CLIP(ResNet-50)| 98.30 | 0.02 |
>
> This finding further validates that higher accuracy in predicting pseudo-spurious labels corresponds to a smaller gap between TIE and TIE*. This motivates our research into methods of enhancing TIE*'s performance. We propose three approaches to enhance performance: employing group-robust text prompts (as discussed in Section 4.3), using a small proportion of labeled data to accurately assign pseudo-spurious labels (as discussed in Section 4.4), and following text prompt guidelines to improve the performance of spurious text prompts (as discussed in A1 of the General Response). These three methods can effectively reduce the performance gap between TIE and TIE*.

---

> ### Author Response · Authors · 2024-11-21
> **part 2**
>
> >(2) In large-scale datasets and tasks, how valid is the assumption of a single spurious feature for the entire dataset? How can spurious features be efficiently identified on a per-data-point basis?
>
> **Q2.** This work builds upon existing research on spurious correlation [1,2,3,4]. Spurious correlation is a statistical concept that describes a relationship between two variables that appears to be related but is actually coincidental or influenced by an external variable [5]. Due to the statistical nature of spurious correlations, identifying spurious features on a per-data-point basis is challenging to achieve for now. This limitation is also noted in Section I of the appendix for further discussion.
>
> For a comprehensive comparison, we conducted experiments across major benchmark datasets commonly used in spurious correlation studies. Our proposed method is capable of addressing multiple spurious correlations simultaneously by utilizing corresponding spurious text prompts, thus extending beyond a single-feature scenario. Additionally, our method supports multi-class spurious labels. For example, in the FMOW dataset,  satellite images from diverse geographical regions show multi-class spurious features. Both TIE and TIE* have addressed multi-class spurious correlations in large-scale dataset studies.
>
> [1] Wu, Shirley, et al. "Discover and cure: Concept-aware mitigation of spurious correlation." International Conference on Machine Learning. PMLR, 2023.
>
> [2] Sagawa, Shiori, et al. "Distributionally robust neural networks for group shifts: On the importance of regularization for worst-case generalization." arXiv preprint arXiv:1911.08731 (2019).
>
> [3] Adila, Dyah, et al. "Zero-Shot Robustification of Zero-Shot Models." The Twelfth International Conference on Learning Representations. 2024.
>
> [4] Kirichenko, Polina, Pavel Izmailov, and Andrew Gordon Wilson. "Last Layer Re-Training is Sufficient for Robustness to Spurious Correlations." The Eleventh International Conference on Learning Representations.
>
> [5] Ye, Wenqian, et al. "Spurious correlations in machine learning: A survey." arXiv preprint arXiv:2402.12715 (2024).

---

### Author Response · Authors · 2024-11-21
**General response part 1**

We sincerely thank all the reviewers for their inspiring and constructive feedback! We deeply appreciate the time and effort spent in providing valuable advice, suggestions, and questions to help refine our submission. We would like to take this opportunity to address the common questions raised.

Reviews **RFvW** and **gGse** share a similar question regarding the guidelines for selecting spurious text prompts:
> "How sensitive is TIE to the choice of spurious prompts? Are there any guidelines for selecting effective prompts? "
> "How do we ensure the accuracy of the textual hints provided when using artificially identified fake features?"


**A1**: One of the major advantages of our proposed method is its robustness. Our method is less sensitive to differences in spurious text prompts compared to existing state-of-the-art method. This robustness is theoretically analyzed in Section 3.4. In addition, we tested 261 different spurious prompts, with the results detailed in Appendix C.1 , see Figures 6 and 7. We also test on different spurious text prompts templates, the results are shown in Table 9. Our observations confirm that our approach is markedly more robust, consistently achieving high worst group accuracy.

Nevertheless, the effectiveness of VLMs still depends on the quality of text prompts. The guidelines for selecting text prompts represent a critical area for deeper exploration. To address this, we show our insights through experiments designed to identify an effective and generalizable approach for creating optimal text prompts in practice.

We investigate this issue by decomposing a text prompt into a *template* and an *object*. Templates include phrases like 'A photo with [Object]', 'A photo with a spurious feature, [Object]', and simply '[Object]'. For the object, [1] shows that labels exhibit a hierarchical structure in WordNet [2].  For example, the hierarchical progression of the word 'strawberry' includes 'berry', 'edible fruit', 'food', each level becoming more general [2]. In our experiments, we test three labeling strategies: using the level directly above to represent a more generalized category, the spurious feature itself, and an average of the top five most specific terms at the bottom of the hierarchy for greater specificity. The aim is to determine the most effective level of generality or specificity for descriptions. Our experiments on the Waterbirds dataset using TIE* (ViT-L14) explore these concepts, with detailed paired text prompts provided below. We selected ViT-L14 to potentially achieve the best performance.

Table 1. Template candidates
||Template|
|-|-|
|T1| A photo with [Object]  |
|T2 |A photo with a spurious feature, [Object] |
|T3| [Object] |


Table 2. Object candidates
||water background |land background |
|-|-|-|
|O1 (hypernyms)| Fluid  | Ground  |
|O2 (self)| Water |  Land  |
|O3 (hyponyms)| Sea, Lake, River , Stream, Creek | Arable Land,  Farmland,  Forest Land, Grassland , Desert |

Table 3. Performance evaluation of CLIP-ViTL14 for TIE*, We have highlighted in bold the results that surpass the TIE* in our submission.
||WGA |ACC | Gap|
|-|-|-|-|
|T1+O1| 53.97 | 76.49  | 22.52 |
|T1+O2| 61.60 | 78.98   | 17.38 |
|T1+O3| **65.26** | 80.20 | 14.94|
|T2+O1| 46.48 |  72.69 | 26.21|
|T2+O2| **63.77** | 80.35   |16.58 |
|T2+O3| **63.19** | 79.06 | 15.87 |
|T3+O1| 45.90 | 73.19  | 27.29 |
|T3+O2| 60.62 | 78.84   | 18.22|
|T3+O3| 59.56 |  77.91 | 18.35 |

**Insights:** We note that using a proper *object* description is important. We suggest using a specific description of the spurious feature or their hyponyms, as this can improve WGA in the zero-shot classification. In contrast, using overly general descriptions such as hypernyms significantly degrades performance. This observation aligns with recommendations for specificity and clarity in text prompt engineering for language models [3].

In terms of templates, we found that giving a portion of contextual information, such as the prefix "a photo with" or "a photo with a spurious feature," helps the WGA. Templates lacking a prefix demonstrate poor performance, a finding that aligns with the observations presented in [4]. For practical purposes in ViT-based CLIP models, we encourage users to adopt templates that include a prefix, with the object description utilizing the spurious feature itself, balancing ease of use and performance.

---

### Author Response · Authors · 2024-11-21
**General response part 2**

Reviewer **y3Qc**, **RFvW** share a similar question regarding the failure case discussion.
> When does it fail?
> What are the failure cases of TIE* and how can automatic inference of spurious labels be improved?

**A2** TIE* is a method free from using any annotations and requires the spurious text prompt for inference of the spurious label in the dataset. We analyzed TIE* failure cases, which can be broadly categorized into two scenarios: (1) inaccuracies in the pseudo-spurious labels and (2) images containing artifacts (e.g., watermarks).



For (1): The majority of failures of TIE* arise when zero-shot classification incorrectly assigns the spurious label. This misassignment leads samples to be translated in the opposite direction, resulting in incorrect classifications. The pseudo-spurious labels assigned by TIE* have a direct effect on the worst group accuracy in zero-shot classification. We substantiate this finding in Section 4.4, as well as in our response Q1 to Reviewer Y3Qc, to which we kindly refer our reviewers for further details. In summary, accurately assigning pseudo-spurious labels enhances TIE*'s performance on the worst group accuracy in zero-shot classification.

To potentially improve TIE*'s performance, we propose three practical strategies: utilizing group-robustified spurious text prompts (as discussed in Section 4.3), employing a small subset of spurious-labeled data (as discussed in Section 4.4), and following the guidelines for effective text prompts (as detailed in A1) to achieve optimal performance.

For (2),  we discussed this scenario in Section I of the appendix. This is a case where the artifact (e.g., a watermark) becomes the dominant feature instead of the spurious feature. While using TIE or TIE* may reduce dependency on spurious features (such as background information), it cannot eliminate the effect of the artifact. This limitation can lead to the failure of our algorithm. Interestingly, we found that the TIE can potentially remove such unwanted features if provides proper text prompts. However, the identification of these incorrect features remains an area under investigation.

[1]Ge, Yunhao, et al. "Improving zero-shot generalization and robustness of multi-modal models." Proceedings of the IEEE/CVF conference on computer vision and pattern recognition. 2023.

[2]Fellbaum, Christiane. "WordNet: An electronic lexical database." MIT Press google schola 2 (1998): 678-686.

[3] Ekin, Sabit. "Prompt engineering for ChatGPT: a quick guide to techniques, tips, and best practices." Authorea Preprints (2023).

[4] Radford, Alec, et al. "Learning transferable visual models from natural language supervision." International conference on machine learning. PMLR, 2021.

---

### Author Response · Authors · 2024-11-25
**Revised version released**

We sincerely thank the reviewers for their constructive suggestions, which have significantly helped improve our work. We have revised the manuscript accordingly, with the major revisions highlighted in blue. A summary of the key revisions is as follows:

(1) Based on Reviewer **wW81**'s feedback, we have revised the notations throughout the manuscript to ensure consistency.

(2) As recommended by Reviewers **RFvW** and **gGse**, we have added a section in Appendix I to provide guidelines for selecting text prompts.

(3) Following the suggestions from Reviewers **y3Qc** and **RFvW**, we have included a section in Appendix K to thoughtfully discuss the failure cases of our methods.

(4) Following Reviewer **wW81**'s suggestion, we explicitly state the requirements for implementing TIE*.


We hope these revisions address the suggestions and questions raised by the reviewers. If there are any further inquiries or clarifications needed,   please do not hesitate to let us know, and we will be happy to provide additional details. Thank you once again for your valuable feedback and effort!

---

### Meta-Review · Area_Chair_1qoG · 2024-12-19

**Metareview:**

The authors propose a new approach to addressing spurious correlations in VLMs in a zero-shot setting, which offers a theoretical grounding and an efficient solution using a translation operation. The method demonstrates robust performance across four image domains, and the authors provide theoretical analysis and visualizations, supporting its generalizability. The paper is well-structured and clearly presented, with relevant insights into the potential improvements for VLMs.

During the rebuttal period, the authors actively engaged in discussions with the reviewers, and the final scores were (8, 6, 6, 6). All reviewers rated the paper positively.  Overall, this paper makes a valuable contribution to addressing spurious correlations in VLMs, and I recommend it for acceptance at ICLR. The authors are encouraged to polish the writing, revise the notations, and discuss more about the guidelines for selecting text prompts and failure cases in the final version.

**Additional Comments On Reviewer Discussion:**

The paper receives all positive scores,  with one strong positive (8) and three borderline accept (6, 6, 6).

Reviewer y3Qc and wW81 raised some concerns about the writing and notations.

Reviewer RFvW and gGse have concerns about the guidelines for selecting text prompts.

Reviewers y3Qc and RFvW recommended the authors discuss more about the failure cases.

After the rebuttal period, the authors addressed all reviewers' concerns and fixed these problems in the paper. Reviewers have no further questions. AC agrees with the reviewers on the novelty and the importance of mitigating spurious correlations in VLMs. I recommend it for acceptance.

---

### Decision · Program_Chairs · 2025-01-22

Accept (Poster)